# Effects of In Utero EtOH Exposure on 18S Ribosomal RNA Processing: Contribution to Fetal Alcohol Spectrum Disorder

**DOI:** 10.3390/ijms241813714

**Published:** 2023-09-05

**Authors:** Nune Darbinian, Gary L. Gallia, Armine Darbinyan, Ekaterina Vadachkoria, Nana Merabova, Amos Moore, Laura Goetzl, Shohreh Amini, Michael E. Selzer

**Affiliations:** 1Center for Neural Repair and Rehabilitation Shriners Hospitals Pediatric Research Center, Lewis Katz School of Medicine, Temple University, Philadelphia, PA 19140, USA; vadacha1@gmail.com (E.V.); nmerabova@gmail.com (N.M.); srsmamos@gmail.com (A.M.); 2Department of Neurosurgery, Johns Hopkins Hospital, Baltimore, MD 21287, USA; ggallia1@jhmi.edu; 3Department of Pathology, Yale University School of Medicine, New Haven, CT 06520, USA; armine.darbinyan@yale.edu; 4Medical College of Wisconsin-Prevea Health, Green Bay, WI 54304, USA; 5Department of Obstetrics & Gynecology, University of Texas, Houston, TX 77030, USA; laura.goetzl@uth.tmc.edu; 6Department of Biology, College of Science and Technology, Temple University, Philadelphia, PA 19122, USA; ashohreh@temple.edu; 7Departments of Neurology and Neural Sciences, Lewis Katz School of Medicine at Temple University, Philadelphia, PA 19140, USA

**Keywords:** FASD, brain development, 18S Ribosomal RNA, alcohol, Ribosomal RNA processing, srRNA, neurons, exosomes

## Abstract

Fetal alcohol spectrum disorders (FASD) are leading causes of neurodevelopmental disability. The mechanisms by which alcohol (EtOH) disrupts fetal brain development are incompletely understood, as are the genetic factors that modify individual vulnerability. Because the phenotype abnormalities of FASD are so varied and widespread, we investigated whether fetal exposure to EtOH disrupts ribosome biogenesis and the processing of pre-ribosomal RNAs and ribosome assembly, by determining the effect of exposure to EtOH on the developmental expression of 18S rRNA and its cleaved forms, members of a novel class of short non-coding RNAs (srRNAs). In vitro neuronal cultures and fetal brains (11–22 weeks) were collected according to an IRB-approved protocol. Twenty EtOH-exposed brains from the first and second trimester were compared with ten unexposed controls matched for gestational age and fetal gender. Twenty fetal-brain-derived exosomes (FB-Es) were isolated from matching maternal blood. RNA was isolated using Qiagen RNA isolation kits. Fetal brain srRNA expression was quantified by ddPCR. srRNAs were expressed in the human brain and FB-Es during fetal development. EtOH exposure slightly decreased srRNA expression (1.1-fold; *p* = 0.03). Addition of srRNAs to in vitro neuronal cultures inhibited EtOH-induced caspase-3 activation (1.6-fold, *p* = 0.002) and increased cell survival (4.7%, *p* = 0.034). The addition of exogenous srRNAs reversed the EtOH-mediated downregulation of srRNAs (2-fold, *p* = 0.002). EtOH exposure suppressed expression of srRNAs in the developing brain, increased activity of caspase-3, and inhibited neuronal survival. Exogenous srRNAs reversed this effect, possibly by stabilizing endogenous srRNAs, or by increasing the association of cellular proteins with srRNAs, modifying gene transcription. Finally, the reduction in 18S rRNA levels correlated closely with the reduction in fetal eye diameter, an anatomical hallmark of FASD. The findings suggest a potential mechanism for EtOH-mediated neurotoxicity via alterations in 18S rRNA processing and the use of FB-Es for early diagnosis of FASD. Ribosome biogenesis may be a novel target to ameliorate FASD in utero or after birth. These findings are consistent with observations that gene–environment interactions contribute to FASD vulnerability.

## 1. Introduction

Fetal alcohol spectrum disorders. Fetal exposure to alcohol during pregnancy is the leading cause of congenital cognitive impairment in the US [1,2,3,4,5,6,7]. Growth restriction, craniofacial abnormalities, deficient brain growth with abnormal morphogenesis, and neurobehavioral impairment are diagnostic criteria for fetal alcohol spectrum disorders (FASD) [7,8,9]. Previous studies of the effects of ethanol (EtOH) on the developing central nervous system (CNS) have emphasized reduced neurogenesis and the induction of neuronal apoptosis [10,11,12]. There is limited knowledge about the mechanisms underlying alcohol toxicity in the developing brain, but the diffuse nature of somatic and behavioral abnormalities in FASD suggests a possible abnormality in the fundamental mechanisms of gene expression. A few studies using mouse and chicken models have identified common gene clusters that are important for neural crest development in EtOH-induced craniofacial dysmorphology [13,14,15,16]. These clusters include ribosome biogenesis and RNA splicing, since early neural crest is sensitive to disruptions in ribosome biogenesis, probably due to the rapid cell proliferation at these early embryonic stages. Interestingly, human ribosomopathies, such as Diamond–Blackfan anemia and Treacher–Collins Syndrome, display craniofacial dysmorphologies similar to those of FASD, and cause cell cycle arrest and apoptosis within the cranial neural crest [17,18]. EtOH strongly suppressed ribosome protein expression as early as 4–6 h after EtOH exposure in mouse and chick headfolds, which are enriched in neural crest [14,15,16].

Effects of prenatal alcohol exposure on ribosomal biogenesis and ribosomal RNAs. Studies on the changes in ribosomal proteins and ribosomal RNAs relative to EtOH-induced cell death are ongoing. Among ribosomal RNAs, 18S ribosomal RNA was shown to be actively involved in interactions between cellular proteins or cell–virus association. Ribosomal biogenesis is an emerging target of neurodevelopmental pathologies [19] and cell growth [20]. Some rRNA processing and translation in mammalian cells was assayed using a synthetic 18S rRNA expression system [21]. Bioinformatic analysis predicted that EtOH exposure during early development would cause alternative splicing alterations of genes involved in RNA post-transcriptional regulation [22]. 

Novel small non-coding ribosomal RNAs. Among non-coding RNAs, long and short rRNAs are novel targets in neurodevelopmental pathologies, and recent reports have focused on mechanisms of fragmentation of rRNAs into short RNAs [23], or on the characterization of small RNAs derived from transfer RNAs (tRNAs), ribosomal RNAs (rRNAs), and small nucleolar RNAs (snoRNAs) [24], or derived from eukaryotic ribosomal RNA [25]. While 18S rRNA quantitation previously was used mostly for normalizing real-time RT-PCR expression analysis, or northern-blot assays [26], recent studies are focused on the new promises and challenges of rRNAs, along with short siRNAs and miRNA short ribosomal RNAs, in health and diseases [27]. Thus, short rRNAs recently have attracted the attention of researchers, while most attention previously had been focused on miRNA maturation or on pre-ribosomal RNA processing. Interestingly, most isolates of small RNAs from cells usually are fragments of rRNAs. Whether processed ribosomal RNAs transform into physiologically relevant small RNAs remains to be established. Probably, rRNAs may also be precursors for functional small miRNA-like moieties. Perhaps, in addition to siRNAs and miRNAs, there is a third class of biologically active small ribosomal RNAs (srRNAs) [27]. Previously, we isolated four 18S-homologous RNAs from human cells, and reported them into GenBank [28,29]. Those four ribosomal RNAs were found in the complex with a cellular protein, Purα, which was previously identified complexed with the myelin basic protein (MBP) gene and controlling its transcription. MBP was recently reported as a potential biomarker for FASD, and regulation of its expression is important. Interestingly, while the cellular protein Purα is in the complex with srRNAs, it also associates with the MBP gene. Furthermore, those rRNAs were essential for binding of Purα to the viral protein Tat during viral infection or treatment of cells with Tat [29]. We also isolated shorter forms, i.e., srRNAs, to study their importance in FASD. A limitation in studies of human FASD is that most EtOH-exposed fetuses do not develop FASD. Therefore, it is difficult to know which at-risk fetus will be born with FASD, and this limits our ability to develop therapeutic interventions. Recently, we found that prenatal exposure to EtOH induces neuronal and oligodendrocyte injury in human fetal brain and eye tissues in vivo, as well as in primary cell cultures derived from human fetuses [30,31,32]. This raises the question of whether there is a novel class of small 18S-homologous rRNAs that is affected by prenatal EtOH exposure. We hypothesize that prenatal EtOH exposure affects biogenesis of 18S-rRNA, which, as we previously demonstrated, is bound to the cellular protein Purα, which in turn controls synthesis of MBP protein, a potential early biomarker for FASD. To explore this hypothesis, we have studied the effect of EtOH on 18S rRNA and its shorter forms in human fetal brain, primary neurons, and fetal brain-derived exosomes (FB-Es) isolated non-invasively from maternal blood. 

## 2. Results

### 2.1. Prenatal EtOH Exposure Inhibits 18S rRNA Expression in Fetal Neurons and FB-Es

We previously organized a human tissue bank of alcohol-exposed fetuses and maternal blood throughout a pregnancy to study the effects of prenatal alcohol (EtOH) exposure on the fetal brain and eye development in humans [31,32,33,34], and to confirm our preliminary data from the in vivo animal model of prenatal alcohol exposure [34]. To study in utero effects of alcohol on 18S rRNA, we used brain tissues and maternal blood from 11–21 weeks gestational age (GA). Clinical characteristics of subjects who consumed EtOH during pregnancy are shown in Table 1. 

We analyzed 18S rRNA as a marker for oxidative injury because the level of 18S rRNA previously was reported to be reduced as markers of oxidative damage in eyes increased [35,36]. Thus, oxidative damage often is accompanied by changes in 18S rRNA expression. Prenatal EtOH exposure was associated with a downregulation of 18S RNA expression in the brain during both the first and second trimesters (220 copies/μL vs. 290 copies/μL in first trimester, or 600 copies/μL vs. 710 copies/μL in second trimester; Figure 1A). A similar pattern, or even stronger downregulation, of 18S rRNA was seen in FB-Es (280 copies/μL vs. 390 copies/μL in first trimester, or 510 copies/μL vs. 650 copies/μL in second trimester; Figure 1B). Thus, downregulation of 18S rRNA expression was greatest in all EtOH cases. These data suggest that FB-E srRNAs levels may prove to be a good marker for predicting whether a child will be born with FAS, even as early as 11 weeks GA.

### 2.2. Generation of 18S-Homologous Small Ribosomal RNAs (srRNAs)

Previously, we isolated four small ribosomal 18S-homologous RNA molecules from human cells [28,29] and mapped them within 18S rRNA (Figure 2A). The four srRNAs contained multiple GC-rich regions, and were strongly associated with the GC repeat rich DNA- and RNA-binding cellular protein (Purα) [37]. We also generated short oligo-srRNA fragments within the 18S-rRNA (Figure 2B). Bioinformatic analysis predicts a secondary structure of srRNA, indicating a stem-loop organization, which possibly is processed by the cellular protein Purα (Figure 2C).

### 2.3. srRNA Reverses Inhibitory Effect of EtOH Exposure on 18S rRNA Expression in Human Neuronal Cells

Next, we checked the biological roles of srRNAs, and whether the addition of srRNAs can reverse the inhibitory effects of EtOH on 18S rRNA expression in neuronal cells treated with 50 mM of EtOH and incubated with srRNA. We found that the addition of srRNAs on EtOH-treated cells resulted in the increase in 18S rRNA expression (1200 copies/mL vs. 600 copies/mL; Figure 3A) measured by ddPCR. This was accompanied by a similar pattern of rescuing effect of srRNA on apoptotic events, on active caspase-3 protein level (50%-increase; Figure 3B), as measured by GLO Caspase-3/7 assay, or on cell survival measured by GLO cell viability assay (10%-increase; Figure 3C).

### 2.4. 18S Ribosomal RNA in Polyribosomes Is in the Complex with the Cellular Protein (Purα)

To characterize novel srRNAs, we first confirmed the presence of srRNA-associated protein, Purα, in ribosomal fractions, using EMSA and ribosome analysis [38]. We demonstrated that ribosome fractions-1-2, 4-7 contain cellular protein Purα in polyribosomal complexes (Figure 4A), while strong srRNA-Purα association was found in polysomes (Figure 4B, lanes 4–6) compared to free ribosomal fractions (lanes 8–10). srRNA contained GGC-rich oligonucleotide [5′-agcttggaggcgga ggcggcctcggcg-3′] corresponding to the wild-type putative Purα-binding site (GGN repeats). 

### 2.5. 18S Ribosomal RNA Is Cleaved by a Cellular Protein (Purα)

To understand the biochemical features of srRNAs, and the formation of srRNAs and their cleavage from the long 18S rRNA, we performed a series of biochemical assays. First, we demonstrated that srRNA-associated Purα exhibits nuclease activity on in vitro-transcribed 18S ribosomal RNA, indicating that one of the mechanisms of formation of smRNA is possibly their cleavage by cellular protein, Purα. Thus, a sequence- and a dose-specific cleavage (lanes 4–5, 6–7) of in vitro-transcribed srRNA by Purα and its deletion mutants (lanes 12–15) was presented in Figure 5A. Furthermore, srRNA was not only nucleased by a novel activity (nuclease activity) of Pura, but it was helix-unwinded first, and then nucleased by Purα, using the partially ds substrate (Figure 5B). Interestingly, a nuclease activity of Purα did not depend on the length of the probe (Figure 5C), in the helix-destabilizing and nuclease assays using the partially ds substrate. The displaced 18-mer and the annealed substrate indicated nucleased cleaved srRNAs.

### 2.6. 18S Homologous srRNA Regulates Cell–Virus Association

To understand the importance of srRNAs in cell functions, we performed in vitro assays to study the association of cellular proteins with other environmental components, in particular, with viral proteins. For example, we demonstrated that a cellular protein, Purα, and a viral protein, HIV-1 Tat, associate with each other via srRNAs (Figure 6A). In vitro-translated Purα (lane 4) was selectively pretreated with agarose-RNase beads before incubation with Tat, then was incubated with GST-Tat in the presence of srRNA-14 (338-nt in sense orientation, lane 5), or with antisense srRNA-14 (338-nt, lane 6), sense srRNA-47 (lane 7), or antisense srRNA-47 (lane 8). Lane 9 contains 1/10 of the amount of [35S]labeled Purα used in the binding reactions, pretreated with RNase. Furthermore, while Purα is unable to associate with Tat when it is treated with RNase, srRNA oligo prevents dissociation of RNase-treated Purα-Tat interaction (Figure 6B). In vitro-translated Purα after RNase treatment was incubated with GST-Tat in the presence of the synthetic oligo-srRNA representing the consensus GC-rich sequence (CCCGGCCGGU) (lane 5) or its mutant variant (AUGACUUGUC) (lane 6), or with the longer form, sense srRNA-14 (lane 7), or AS srRNA-338 (lane 8). In addition, 18S srRNA oligo is specific for Tat-Purα association; while it is not important for the interaction of Pura with another cellular partner, E2F1 (Figure 6C), although Purα-E2F1 association is also RNA-mediated [39,40,41], but another type of RNA is probably essential to restore the Purα-E2F1 complex. Binding reactions are with in vitro-translated Purα after RNase treatment (as indicated) and GST-E2F, in the presence of the in vitro-transcribed srRNA-338 fragment in the sense orientation (lane 5). The functional importance of 18S homologous srRNA-mediated Purα-Tat association was studied in reporter assays, demonstrating a strong inhibition of Tat-activated HIV-1 LTR transcription activity in human neuronal cells (bar 6) and abrogation of Purα-induced or Tat- and Purα-induced HIV-1 LTR activation (bars 7–8, Figure 6D), and indicating the possible use of exogenously added srRNAs in the inhibition of HIV-1 gene transcription. Functional importance of Purα-associated RNA in Tat and Purα effects on LTR activity. Human primary neuronal cells were transfected with 0.5 μg of LTR-luciferase reporter plasmid in the absence or presence of 1 μg of expression plasmids for Tat, Purα, and the srRNA-338 alone or in combination, as indicated at the bottom of the graph. Luciferase activity was determined 36 h after transfection, and the levels of promoter activation by Tat, Purα, and srRNA-338 were determined and graphed as fold activation. In all cases, the final amount of DNA in the transfection mixture was brought to 3.5 μg with pCMV DNA.

In summary, in vitro-translated 35S labeled Purα (lane 1) was incubated with GST (lane 2) or GST-Tat protein (lane 3 in Figure 6A,B) or another cell protein E2F1 (lane 3 in Figure 6C) for 1 h at 4 °C. The complex was then pelleted, washed with binding buffer, eluted by boiling in Laemmli sample buffer, and separated by SDS/PAGE. Proteins were detected by autoradiography. An amount equivalent to 10% of the material used for the assays is shown in lane 1. 35S labeled in vitro translated Purα was treated with RNase and then incubated with GST-Tat (lane 4 in Figure 6A,B), or GST-E2F1 (lane 4 in Figure 6C). Reconstitution of the interaction between Purα and Tat protein was performed by the addition of srRNA (lanes 5–8). 

### 2.7. Potential Targets of srRNA

To understand whether srRNAs play roles in another disorders, or may have another associations or targets, we performed a bioinformatic blast analysis of srRNAs and demonstrated the distribution of 1570 genes homologous to 19-nucleotide srRNA, which is involved also in Alzheimer’s disease (Figure 7A). The distribution of another 43 genes homologous to the longer, 338-nucleotide srRNA is shown in Figure 7B,C. Our data revealed a potential mechanism of srRNA processing by a cellular protein Purα (Figure 7C). Thus, by studying srRNAs, we found novel activities of srRNA-associated cellular protein Purα, which can be a novel member of Class IV RNase III (Figure 7D), summarized in previous and current articles (Figure 7E), which include protein–protein interaction via RNA, DNA- and RNA-binding, binding to ribosomes, DNA-unwinding [42], novel DNase and RNase nuclease activities, possible phosphatase activity, control of transcription/translation, and involvement in DNA-damage repair. Most Purα interactions are RNA-dependent.

### 2.8. EtOH-Induced Reduction in FB-E srRNA Content Correlates with Reduction in Human Fetal Eye Size

Not every child exposed to EtOH in utero develops FASD. In the present study, pregnancies were terminated. Thus, we have no clinical follow-up with which to assess the predictive value of specific molecular markers in predicting whether a fetus will go on to develop FASD postnatally. However, we can examine the fetuses for anatomical anomalies that are characteristic of FASD. One such anatomical hallmark that is easy to measure in the fetus, even after pregnancy termination, is eye size (because eyes are so small, they are almost always intact). We previously demonstrated a negative correlation between EtOH use by pregnant mothers and several parameters of fetal development, including smaller eye sizes compared to controls [32,34]. We now demonstrate a high correlation between EtOH-induced reduction in FB-E srRNA content and a reduction in fetal eye size. Eye diameters were measured in histological sections as previously described [32], while srRNA levels were measured by ddPCR (for copies/μL). Each EtOH-exposed fetus was compared with its GA-matched control (Figure 8). Thus, reduced srRNA levels in FB-E accurately predicted an anatomical hallmark of FASD, and may eventually prove to be predictive of FASD in pregnancies allowed to come to term.

## 3. Discussion

The present study is significant in two ways. First, it establishes small ribosomal RNAs as an important cellular feature of the effects of fetal alcohol exposure that can be detected even in the first trimester. Second, the study identifies molecular biomarkers, particularly several small forms of 18S rRNAs, that can be quantified in fetal-derived exosomes isolated non-invasively from maternal blood, processed by to a novel activity of cellular protein Purα, nuclease activity, never reported previously. These two findings may allow us to conduct large-scale population-based studies to determine whether these molecular markers can predict the emergence of FASD. 

### 3.1. Small Ribosomal RNAs as a New Indicator of Alcohol-Associated Fetal Pathology

The effects of prenatal alcohol exposure on fetal development are complex and often severe. Congenital abnormalities include prenatal and postnatal growth retardation, CNS abnormalities including developmental delay and intellectual impairment, and facial abnormalities [3,46,47,48]. The effects of prenatal alcohol exposure on early fetal development are difficult to determine in utero using current imaging technology. Thus, our study selected a biomarker that could be detected early in fetuses during the vulnerable developmental period between three and six weeks after fertilization [49,50], when EtOH can damage the cranial neural crest cells that are critical for development of facial features, and when rRNAs are involved in a marked reduction of cranial neural crest cell proliferation and survival, impaired migration, and increased apoptosis [19,22]. Here, we performed developmental studies on sr18S-RNA in human brain using ddPCR, and demonstrated: (i) developmental regulation of srRNAs in human fetal brain (Figure 1A) and in FB-Es (Figure 1B); (ii) the effects of in utero EtOH exposure on srRNA in developing brain (Figure 1 and Figure 3); (iii) the association of MBP gene-binding protein, Purα, with 18S ribosomal RNA, which results in its cleavage, with the formation of small RNAs (smRNA) (Figure 2, Figure 4 and Figure 5); (iv) the role of Purα and 18S ribosomal RNA in viral–host interaction (Figure 6); and (v) DNase and RNase nuclease activity and subsequent unwinding activity of the MBP gene-binding Purα (Figure 5 and Figure 7). In an earlier study on HIV, we suggested that in addition to siRNAs and miRNAs, there might be a third class of biologically active small ribosomal RNAs [27]. The present report suggests the existence of such new srRNAs, which previously were identified only in plants, wheat seeds, and zebrafish, and that Purα has activities in addition to the helicase activity that we described previously [42,51,52,53,54].

The mechanism by which EtOH affects RNAs is unknown, but recently, it was suggested that EtOH modifies the alternative splicing of genes related to post-transcriptional regulation, which likely affects neuronal proteome complexity and brain function [22].

### 3.2. Potential Molecular Biomarkers, Using Novel Non-Invasive Tools

The present study has identified several molecular biomarkers belonging to a novel class of non-coding RNAs—small ribosomal RNAs that correlate with neuronal injury and can be detected in FB-E obtained non-invasively from maternal blood samples during early stages of pregnancy. These markers might be useful in predicting significant developmental abnormalities. There was a strong association between early EtOH exposure, GA, ribosomal RNA changes and increased activation of caspase-3, a marker of apoptosis that can be detected before facial features are apparent, in the fetal brains (Figure 1A), fetal neurons (Figure 3), and FB-Es (Figure 1B) exposed to EtOH, suggesting that the developmental abnormalities seen in FASD are due to excessive apoptosis. Previously, we showed that EtOH exposure increased apoptotic signaling (activated caspase-3) in neurons, OL progenitors and mature OLs, suggesting that the dysmyelination seen in FASD was due to defects in both the generation of mature OLs and apoptosis of mature OLs [31,32,33]. Here, we propose a non-invasive technique to determine which fetus would be born with FASD. By analyzing the exosomes in maternal blood, using fetal-specific markers, we isolated a source of fetal brain-specific rRNA biomarkers that can be accessed non-invasively (Figure 1B). The contents of these FB-Es have suggested biomarkers, specifically srRNAs, which are processed by a cellular protein Purα that was previously discovered to be complexed with another biomarker, MBP, and its promoter, and whose levels are significantly lower in the fetuses of women who consumed alcohol during pregnancy. A highly significant relationship between FB-E MBP levels and eye diameter was previously shown [32], suggesting that future large-scale clinical studies might focus on FB-E MBP and its partners, in the search for practical markers to diagnose FASD prenatally [55,56,57].

### 3.3. Formation of srRNAs 

The cellular protein, Purα, which has a strong affinity for single-stranded nucleic acids containing the (GGN)n sequence, exhibits diverse biological activities on RNA transcription, RNA transport and translation, DNA replication, DNA damage, cell cycle, and cell proliferation. Here, we demonstrate that Purα is physically present in polysomes and associates with 18S ribosomal RNA and 18S homologous small ribosomal RNAs (srRNA). The secondary structure of srRNA-338 reveals a 19-nucleotide stem-and loop organization (Figure 2). By associating with srRNA-338, Purα exhibits novel sequence-specific nuclease activity, resulting in the cleavage of srRNA-338. We also demonstrated the importance of srRNAs in Purα-HIV-1-Tat association. Moreover, we showed that srRNA-338 was able to abrogate Purα-mediated inhibition of transcription and translation in vitro and in primary or proliferating cells, where srRNA-338 cooperates with Tat and Purα in a cell-type-specific manner (Figure 6D). Finally, computer blast analysis of smRNA-338 and its 19-nucleotide stem srRNA-19 revealed more than 1600 genes that can be potential targets for Purα-associated srRNAs, emphasizing the importance of the srRNAs in host cell–viral interactions. The association of Purα with 18S rRNA results in the cleavage and processing of small ribosomal RNAs (smRNA). Thus, Purα may belong to a novel class IV of the RNase III family, with the ability to process a novel class of untranslated RNA molecules, srRNA, that are important for Purα-Tat association and can control viral gene transcription, translation, and viral replication.

MBP gene binding by Purα. Purα is a transcription factor implicated in diverse cellular and viral functions, including transcription, replication, and cell growth [51]. In a previous study, we examined biomarkers for myelin development because of the frequency with which periventricular leukomalacia is found in FASD. Reduced MBP expression in particular showed a strong correlation with reduced eye diameter [31,32,33], suggesting that MBP might be a good biomarker to predict FASD. Purα originally was purified from mouse brain based on its ability to bind to the DNA fragment containing a GGCGGA sequence derived from the MBP proximal regulatory region [52,53,54]. Expression of Purα in mouse brain and its DNA binding activity are developmentally regulated and peak at 15 to 18 days after birth [53,54], a time at which many inter-neuronal connections are being established, particularly in the cerebellum. Purα also was found in hypomyelinated brains [58], and we demonstrated that EtOH delays myelination [32] and inhibits MBP expression [33]. While many actions of Purα occur in the nucleus, the protein is frequently located in the cytoplasm, depending at least in part upon the cell cycle phase. Purα can interact with the regulatory proteins of several viruses, including HIV-1 Tat, which is required for viral replication in infected CNS cells, and this association is RNA-dependent.

Purα–Tat association via srRNAs. Purα binds to Tat and the Tat–Purα interaction is mediated by srRNAs immunopurified from human cells [28,29]. Purα interacts with at least five RNA species homologous to 18S rRNA and inhibits translation in vitro. Recent studies indicate that Purα participates in dendritic transport of mRNAs and associates with ribosomes [59]. Unique features and functions of Purα raise the possibility that Purα may be involved in regulation of cellular MBP or viral HIV gene expression. Purα may mediate the association of mRNPs with polyribosomes via binding to rRNA. Interestingly, our earlier studies revealed that Purα binds to RNAs that are homologous to 18S ribosomal RNA and to 7SL RNA, and that association of Purα with RNAs homologous to 7 SL determines its binding ability to the MBP promoter DNA sequence [28,29,60], demonstrating a potential mechanism of regulation of MBP gene expression, and indicating the importance of srRNAs-Purα association and regulation of MBP also in FASD. 

Small Ribosomal RNA as novel early biomarkers for FASD. Because reduced MBP expression proved to be so closely correlated with small eye diameter, we suggested that it might serve as a biomarker to predict the emergence of FASD postnatally. Therefore, in the present study, we looked for a similar correlation between ribosomal RNAs in FB-Es to determine whether we could find even more sensitive predictors of FASD in at-risk infants (Figure 8). The present findings suggest that inhibition of srRNAs might be particularly promising because of the very strong correlation between the increase in srRNA damage and the magnitude of one of the anatomical phenotypic hallmarks of FASD, i.e., reduced eye size [3]. Large-scale prospective studies in pregnancies brought to term could test the diagnostic value of FB-E cargos, including those relating to srRNA, in predicting the emergence of FASD postnatally.

Limitations. Because of the large variabilities in human studies, the present study will have to be expanded to larger sample sizes to allow for more precise controls for such factors as fetal sex, ethnicity, racial groups, use of other medications or substances of abuse, maternal obesity, and maternal psychiatric conditions. In addition, the present study used tissue bank material that was accumulated specifically for the study of FASD, but without follow-up studies. Importantly, although we were able to confirm most of our FB-E studies in blood with our data using fetal brain tissues, we are planning follow-up studies to determine the predictive value of the biomarkers we have studied in identifying which at-risk fetuses would go on to have FASD. 

The quantification of EtOH consumption was based on the subjects’ information in face-to-face interviews, not on biochemical assays, e.g., blood alcohol levels or hair ethyl glucuronide (EtG) levels. Because such tests remain positive only for a short time after cessation of alcohol exposure (blood and urine EtG levels detect alcohol use only for the previous few days to weeks) we were not able to test for alcohol use at the time of termination, since these tests could not rule out early prenatal use. However, in other studies, we used a questionnaire to exclude exposure to SSRIs, opioids, marijuana and other medications and drugs, and we were confident about most of the subjects’ information in this patient population, because in those studies, the regular use of other drugs was confirmed by biochemical assays. 

## 4. Methods

### 4.1. Clinical Samples 

We used 20 EtOH-exposed human fetuses and 20 unexposed controls matched by gestation age, selected from among a total of 153 EtOH-exposed and 71 unexposed control elective pregnancy terminations, in which none of the pregnant women used drugs or medications [31,32,33,34]. The selection of cases and controls was made by the availability of matching maternal blood samples and intact fetal eye and brain tissues, and the availability of data for matching of sex, ethnicity/race, and gestational age (GA). Consenting mothers were enrolled between 11 to 21 weeks GA under a protocol approved by our Institutional Review Board (IRB). All or an appropriate subset of these fetuses were analyzed, matching each EtOH-exposed fetus with its control, at a minimum, with regard to sex and GA. 

Cases (EtOH users) were matched to controls (non-users) by GA and fetal sex (10 males and 10 females), depending on biobank availability. All assays were performed in triplicate. Sex was determined using commercially available SRY primers (Integrated DNA Technologies, Inc., Coralville, IA, USA) and SuperScript One-Step RT-PCR with Platinum Taq (Life Technologies, Carlsbad, CA, USA). Immediately following elective pregnancy termination, surgical tissue samples were collected by our laboratory trained study coordinators; both fresh and snap frozen samples were obtained and transferred to the laboratory within 40–60 min. Several aliquots were either used for RNA and protein extraction or stored in liquid nitrogen for future use. Initial histologic staining of brain tissues from the Biobank confirmed that we had collected mostly cerebral cortex (see 2.10. Immunohistochemistry). Clinical characteristics of participants are summarized in Table 1. Data from both sexes were combined.

Assessment of Alcohol Exposure in Pregnancy: Maternal EtOH exposure was determined with a face-to-face questionnaire that also included questions regarding many types of drugs/medications used, as well as tobacco exposure [31,32,33,34,55,56,57,61]. EtOH exposure was defined as current daily use, and samples were matched based on the last incidence of alcohol consumption, as indicated by the survey. Alcohol dose was calculated as the total number of drinks consumed in a week multiplied by the number of weeks of exposure. Alcohol exposure was assessed using a detailed questionnaire based on measures adapted from the NICHD PASS study [62]. EtOH consumption for each week since conception (2 weeks after last menstrual period) was self-reported using visual/photographic guides of different types of drinks to estimate actual EtOH dose. Women admitting to any EtOH use were classified as EtOH-exposed. Although significant details on EtOH use were collected, our four most important exposure variables for subsequent analysis were current use (Y/N), exposure pattern, cumulative dose, and average weekly dose. Current use is defined as EtOH use in the 5 days prior to enrollment (Yes/No). Exposure pattern was categorized as social (use is not daily and is <4 drinks on any one occasion), binge (>4 drinks on any occasion), or heavy (daily or near daily EtOH use). Cumulative dose is a continuous outcome that is defined as the estimated total ounces of pure alcohol consumed since conception. Average weekly dose is a continuous outcome defined as cumulative dose divided by the total number of weeks since conception. The total cumulative alcohol dose for EtOH-consuming mothers ranged from 57–168 drinks (or 12–30 drinks/month) in the first trimester, and from 54.4 to 1827.5 drinks (or 6–320 drinks/month) for the second trimester. A drink was estimated as the equivalent of one shot (1.5 oz of brandy or 5 oz of wine [32,34]. Women with a known urinary tract infection or a urinalysis with white blood cells or nitrates were not enrolled. Sensitivity for EtOH use within the last 120 h at this cutoff is only 70% with a specificity of 99% [63]. HPLC-based blood and urine testing for amphetamines and other drugs was performed previously on many of our other subjects, to validate recent drug use in relation to other studies, and we found concordance between reported use and actual use to be greater than 90% in this patient population. Perhaps women who terminate their pregnancy have fewer barriers to truthful reporting than do women in the general population. 

Subject Recruitment: Women reporting alcohol use (or no alcohol use) since conception were grouped from two GA windows: first trimester and second trimester. GA was determined by a dating ultrasound performed immediately prior to recruitment; at the GAs planned, ultrasound can accurately determine GA ± 10 days [64]. 

Sample Collection and Processing: Fetal brain and eye tissues and maternal blood were collected for each subject. Fetal brain tissue from subjects undergoing elective termination of pregnancy was collected according to an IRB-approved protocol. Surgical tissue samples were collected immediately by a trained study coordinator. Both fresh and snap-frozen samples were transferred to the laboratory within 30 min. Then aliquots either were used for RNA extraction or kept in liquid nitrogen for future use. The whole brain/forebrain was used in this study. Initial histologic staining of brain tissues from the Biobank confirmed that we had collected mostly cerebral cortex [31]. Alcohol-exposed or control brain samples were used previously as we indicated [31,32,33,34] to study oligodendrocyte and neuronal markers. Specimens have been banked for up to 10 years after the completion of a study to allow for NIH data and sample sharing. 

We aimed for a final sample of 20 EtOH-exposed and 20 case-matched control fetuses that had the requisite combination of maternal blood samples, maternal race matching, fetal GA and sex matching, and well-preserved fetal anatomical structures. As soon as that number was reached, those 40 pregnancies were incorporated into most of the detailed analyses. Eye globe diameter (lateral), eye length (anterior–posterior), and pupil diameter and shape were noted for both the right and left eyes of all fetuses, and informative comparisons were made.

Ethics. Human subjects. Consenting mothers were enrolled between 11–21 weeks gestation, under a protocol approved by our Institutional Review Board (IRB). This protocol involved no invasive procedures other than routine care. Maternal EtOH exposure was determined with a face-to-face questionnaire that also included questions regarding many types of drugs/medications used [31,32,55,56]. The questionnaire was adapted from that designed to identify and quantify maternal EtOH exposure in the NIH/NIAAA Prenatal Alcohol and SIDS and Stillbirth (PASS) study [56]. 

All procedures for collection and processing of human brain tissues and blood were performed according to NIH Guidelines by a trained study coordinator. All investigators completed Citi Program—Human Subject training, Blood-Borne Pathogens Training, and Biohazard Waste Safety Training annually (see details in [32]). 

Written informed consent was obtained from the parents of patient(s) for studies, and deidentified samples were used for this publication. Informed consent forms were maintained by the study coordinator. The deidentified log sheets contain an assigned accession number, and the age, sex, ethnicity, and race of the patient. Except for an assigned accession number, no identification was kept on the blood samples. 

Eligibility Criteria: The blood and tissue samples were obtained according to NIH Guidelines through a trained study coordinator. Samples were collected regardless of sex, ethnic background, and race.Treatment Plan: Each patient was asked to sign a separate consent form for research on blood and tissue samples. Blood obtained was processed for collection of serum and plasma. No invasive procedures were performed on the mother, other than those used in her routine medical care. Fetal tissues were processed for RNA or protein isolation. Risk and Benefits: There are very small risks of loss of privacy, as with any research study where protected health information is viewed. The samples were depersonalized before they were sent to the lab for analysis. There were no additional risks of blood sampling as this was only performed in patients with clinically indicated venous access. There was little anticipated risk from obtaining approximately 2–3 cc of blood, but a well-trained study coordinator collected all samples.

There was no direct benefit to the research subjects from participation, but there is significant potential benefit for the future FAS subjects and the general population. This research represents a reasonable opportunity to further the understanding, prevention, or alleviation of a serious problem affecting the health or welfare of FAS patients.

d.Informed Consent: Consent forms were maintained by the study coordinator and were not sent to the investigator with the samples. The deidentified log sheets and IRB protocol were sent by the study coordinator to the principal investigator with each blood and tissue sample. This sheet contains an assigned accession number, the age, sex, ethnicity, and race of the patient. Except for an assigned accession number, no identification was kept on the blood and tissue samples.

Consenting mothers were enrolled between 11 and 21 weeks gestation, under a protocol approved by our Institutional Review Board (IRB). This protocol involved no invasive procedures other than routine care. Maternal EtOH exposure was determined with a face-to-face questionnaire that also included questions regarding many types of drugs/medications used, as well as tobacco exposure [31,32,55,56]. The questionnaire was adapted from that designed to identify and quantify maternal EtOH exposure in the NIH/NIAAA Prenatal Alcohol and SIDS and Stillbirth (PASS) study [64]. 

All procedures involving collection and processing of blood and tissues were performed according to NIH Guidelines through a trained study coordinator. All investigators were trained annually to complete Citi Program—Human Subject training, Biohazard Waste Safety Training and Blood–Borne Pathogens Training, as well as all other required training. 

### 4.2. Cell Culture 

Human primary cortical neurons were prepared in our laboratory by Dr. Darbinyan [34,65,66,67,68,69]. Human fetal brains and fetal eyes resulting from elective abortion were obtained from Advanced Bioscience Resources (ABR), Inc., Alameda, CA 94501, USA, under a protocol approved by Temple University’s IRB. The protocol also complied with NIH guidelines at the time. In brief, 16-week old fetal brain (approx. 13 g) was treated with Tryple Express enzyme (Invitrogen, Carlsbad, CA, USA), DNase I (10 U/mL; Sigma, St. Louis, MO, USA) for 15 min and maintained at 37 °C, then washed three times with Hibernate E medium. Neurobasal medium containing B27 supplement and 0.25 mM Glutamax was used for a tissue trituration with a glass Pasteur pipette, and cells were then plated on poly-D-lysine-coated 60 mm dishes (Sigma, St. Louis, MO, USA). After 16 h, 1 μM of Cytosine arabinoside Ara-C (Sigma, St. Louis, MO, USA) was added to cells for 48 h. Cells were cultured in Neurobasal medium containing several antibiotics, including 10 μg/mL gentamycin, 100 units/mE penicillin and 10 μg/mL streptomycin. Medium also contained one μg of antifungal fungizone (Life Technologies, Inc., Carlsbad, CA, USA). Cells were maintained at 37 °C incubator.

Cell Treatment. Neuronal cells were incubated with 50 mM EtOH [66], or with recombinant Tat, rTat 101 (ImmunoDiagnostics, Inc., Woburn, MA, USA), (50 ng/mL, or 3.6 nM), for a total of 48 h [67,69,70,71]. 

### 4.3. RNA Preparation and Real-Time Quantitative Polymerase Chain Reaction (qRT-PCR) 

Human fetal total RNA was isolated using the RNeasy kit (Qiagen, Valencia, CA, USA) with on-column DNA digestion. The RT-PCR reaction was performed with 1 μg total RNA, using the One-Step FAST RT-PCR SYBR Green PCR Master Mix (Qiagen). The StepOnePlus Real-Time PCR system thermo cycler was used (Applied Biosystems, Waltham, MA, USA). PCR conditions were as follows: activation 95 °C 5 min; PCR 45 cycles: 95 °C 10 s, 60 °C 20 s, 72 °C 30 s; melting curve (95–65 °C); cool to 40 °C 30 s. For relative quantification, the expression levels of genes were normalized to the housekeeping gene β-actin [31,32,33,34].

### 4.4. Sex Determination Using Human Fetal Genomic DNA 

Sex determination was carried out using SuperScript One-Step RT-PCR with Platinum Taq (Life Technologies, Carlsbad, CA, USA) and a BioRad C1000 Touch Thermal Cycler. In parallel studies, total cellular genomic DNA was isolated from fetal brain tissue for PCR analysis using the QIAamp DNA isolation kit (Qiagen) and primers for SRY gene. Amplification was performed in a GeneAmp PCR System 2400. Products were visualized by gel electrophoresis using 2% agarose gel and GelRed DNA stain. The thermal cycler program used was 45–55 °C for 15–30 min, 94 °C for 2 min, 55–60 °C for 30 s, 68–72 °C for 1 min, 72 °C for 5–10 min, and 12 °C holding temperature. Sex determination in fetal tissue using RNA: RNA was extracted from fetal tissue and one-step RT-PCR reaction was performed utilizing SRY primers. All PCR reactions were performed in a thermal cycler (C1000 Touch™, BioRad, Hercules, California, USA) at 94 °C (2 min), followed by 35 cycles of 94 °C (15 s), 65 °C (20 s) and 72 °C (20 s), with a final extension of the cycle at 72 °C for 10 min. The amplified PCR products were separated on 2.5% agarose gels, GelRed stained, and visualized under UV transillumination.

### 4.5. Droplet Digital PCR (ddPCR)

For absolute quantitation of mRNA copies, ddPCR was performed using the QX200 ddPCR system. A total of 50 ng of human fetal total RNA was used with the 1st Strand cDNA Synthesis Kit (Qiagen, Germantown, MD, USA). After reverse transcription, the cDNA (300 dilution) aliquots were added to BioRad master mix to conduct ddPCR (EvaGreen ddPCR Supermix, BioRad, Hercules, CA, USA). The prepared ddPCR master mix for each sample (20-μL aliquots) was used for droplet formation. PCR conditions: Activation 95 °C 5 min, PCR 45 cycles at 95 °C 10 s, 60 °C 20 s, 72 °C 30 s, melting curve (95–65 °C), cool to 40 °C 30 s. The absolute quantity of DNA per sample (copies/µL) was calculated using QuantaSoft Analysis Pro Software (Bio-Rad, Hercules, CA, USA) to analyze ddPCR data for technical errors (Poisson errors) [31]. A greater total number of droplets results in higher accuracy. With 20,000 droplets, the above ddPCR protocol yields a linear dynamic range of detection between 1 and 100,000 target mRNA copies/µL. The estimated error is negligible compared with other error sources, e.g., pipetting, sample processing, and biological variation. The ddPCR data were exported to Microsoft EXCEL for further statistical analysis. 

### 4.6. Primers (IDT Inc., Coralville, IA, USA)

β-actin: S: 5′-CTACAATGAGCTGCG TGTGGC-3′, 

AS: 5′-CAGGTCCAGACGCAGGATGGC-3′, 

SRY: Forward 5′-CAT GAA CGC ATT CAT CGT GTG GTC-3′; reverse 5′-CTG CGG GAA GCA AAC TGC AAT TCT T-3′.

srRNA oligonucleotide (22-nt): 5′-AACGAAGGGCACCACCAGGAGT (for nuclease assay)

srRNA 338-nt: 5′-

1 acctcacccg gcccggacac ggacaggatt gacagattga tagctctttc tcgattccgt

61 gggtggtggt gcatggccgt tcttagttgg tggagcgatt tgtctggtta attccgataa

121 cgaacgagac tctggcatgc taactagtta cgcgaccccc gagcggtcgg cgtcccccaa

181 cttcttagag ggacaagtgg cgttcagcca cccgagattg agcaataaca ggtctgtgat

241 gcccttagat gtccgggccg ggtgaggttt cccgtgttga gtcaaattaa gccgcaggct

301 ccactcctgg tggtgccctt ccgtcaattc ctttaagt

Universal 15-mer: 5′-AGTCACGTTGACGTA-3′ (for unwinding assay)

GGN-Purα: 5′-AGCTTGGAGGCGGAGGCGGCCTCGGCG-3′ (for band-shift-assay) [72]

srRNAs were isolated using reverse transcription followed by polymerase chain reaction assays as previously described [28,29]. Clones 10, 11, 14, and 47 were subcloned into pCNDA3 (Invitrogen, Carlsbad, CA, USA). 

pCMV-sr14.4 was created by placing the 338-base cDNA representing the Purα-associated rRNA (GI: 15011542), respectively, at the 3’ end of the CMV promoter in the EcoRI-linearized pCDNA3 vector [28,29].

pCMV-sr14.4 as: 338-base cDNA was cloned in the antisense orientation in pCDNA3 [28,29]. 

pGL3 T7-luciferase plasmid was constructed by digesting pCDNA3-TAR, a vector that contains the HIV-1 TAR sequence cloned into the HindIII/BamHI sites of pCDNA3, with SmaI and BamHI. This SmaI/BamHI fragment, which contains the T7 promoter, and the HIV-1 TAR sequence, was then subcloned into SmaI/BglII digested pGL3-basic vector (Promega, Madison, WI, USA), generating pGL3-T7-TAR. This construct was subsequently digested with HindIII, releasing the TAR fragment, and the vector containing the T7 promoter, and the luciferase gene was re-ligated, generating pGL3-T7-luciferase. The sequence of all plasmids and PARNA clones was verified by DNA sequencing using an ABI automatic sequencer. 

pCMV-Tat was created by placing the Tat cDNA at the 3′ end of the CMV promoter in the pCDNA3 vector.

HIV-1 glutathione S-transferase (GST)–Tat expression vector was obtained from the National Institutes of Health AIDS Research and Reference Reagent Program (Bethesda, MD, USA). 

GST-Pura and its mutant variants have been previously described [42]. 

pEBV-His B-Purα contains the coding region of the Purα gene cloned downstream of a histidine epitope tag. pEBV-His B-Purα was constructed by first subcloning the Purα EcoRI fragment from pGEX1λT-Purα into EcoRI-digested pCDNA3 (Invitrogen), generating pCDNA3-Purα. Subsequently, the BamHI/XhoI fragment from pCDNA3-Purα was cloned into BamHI/XhoI cleaved pEBV-His B (Invitrogen), generating pEBV-His B-Purα. 

### 4.7. Isolation of Fetal Brain-Derived Exosomes (FB-Es) from Maternal Plasma

Human FB-Es were isolated as described previously [32,33,34,55,56]. A total of 250 µL of maternal plasma was used to isolate exosomes. Because our previous studies revealed that EtOH reduced the number of FB-Es, all exosomal assays were normalized against the exosomal marker CD81. After centrifugation, supernatants were incubated with exosome precipitation solution (EXOQ; System Biosciences, Inc., Mountainview, CA, USA). The resultant suspensions were centrifuged at 1500× *g* for 30 min at 4 °C, and pellets resuspended in 400 mL of distilled water with protease and phosphatase inhibitor cocktail for immunochemical enrichment of exosomes. To isolate exosomes from fetal neural sources, total exosome suspensions were incubated for 90 min at 20 °C with 50 μL of 3% bovine serum albumin (BSA) (Thermo Scientific, Inc., Waltham, MA, USA) containing 2 μg of mouse monoclonal IgG1 anti-human contactin-2/TAG1 antibody (clone 372913, R&D Systems, Inc., Minneapolis, MN, USA), that had been biotinylated (EZLink sulfo-NHS-biotin System, Thermo Scientific, Inc.). Next, 10 μL of Streptavidin-Plus UltraLink resin (Pierce, Thermo Scientific, Inc., Waltham, MA, USA) in 40 μL of 3% BSA was added, and the incubation continued for 60 min at 20 °C. After centrifugation at 300× *g* for 10 min at 4 °C and removal of supernatants, pellets were resuspended in 75 μL of 0.05 mol/L glycine-HCl (pH 3.0), incubated at 4 °C for 10 min and recentrifuged at 4000× *g* for 10 min at 4 °C. Each supernatant was mixed in a new 1.5 mL Eppendorf tube with 5 μL of 1 mol/L Tris-HCl (pH 8.0) and 20 μL of 3% BSA, followed by addition of 0.40 mL of mammalian protein extraction reagent (M-PER; Thermo Scientific, Inc.) containing protease and phosphatase inhibitors, prior to storage at −80 °C. 

### 4.8. Ribosome Analysis 

Ribosome analysis was performed as described [38]. Briefly, 1.5 × 10^7^ cells were treated with cycloheximide (CHX; 100 μg/mL) for 10 min and harvested by gentle scraping. The cells will be pelleted, washed with ice-cold PBS containing 100 μg/mL of CHX, pelleted again and then lysed by the addition of 0.5 mL of polysome lysis buffer, containing 0.3 M KCl, 5 mM MgCl_2_, 10 mM HEPES (pH 7.4), 0.5% Nonidet P-40, 100 μg/mL of CHX, 250 μg/mL of heparin, 0.1 U/mL of RNasin, and 10 U/mL of Prime RNase Inhibitor. The lysate will be passed through a 27-gauge needle and centrifuged at 12,000× *g* rpm for 5 min to remove nuclei and cell debris. Aliquots of 20 *A*_260_ units were loaded on 11 mL of 5 to 47% sucrose gradients prepared in a buffer containing 20 mM Tris-HCl (pH 7.4), 10 mM MgCl_2_, 0.5 M KCl, 50 mM NaCl, and were subjected to centrifugation at 39,000 rpm for 3 h in a Beckman SW41 rotor at 4 °C. Gradients were fractionated by pipetting from the top, and about 10 fractions of 0.5 mL each were collected. Each fraction was dialyzed in LB 150 (50 mM Tris-HCl (pH 7.4), 150 mM NaCl, 5 mM EDTA, 50 mM NaF to remove sucrose and salt. We isolated approximately 200 μg of ribosomes and about 100 μg of ribosomal proteins. Approximately 10 μg of ribosomes (25 μL) from each fraction was used in electrophoretic mobility shift assays (EMSA) to demonstrate the presence of Purα in polyribosomal complexes.

### 4.9. Electrophoretic Mobility Shift Assay (EMSA)

Cell lysates were prepared and EMSA was performed as described previously [42]. srRNAs or 27-mer oligonucleotide corresponding to the wild-type putative Purα-binding site (GGN repeats) were synthesized, annealed to complementary synthetic oligonucleotide in vitro, and radiolabeled with [γ-^32^P]-dCTP using Klenow DNA polymerase (New England Biolabs, Beverly, MA, USA). DNA binding reactions were performed in 20 μL containing 10 μg of the nuclear extracts, 2 μL of DNA binding buffer (42), 1 ng of labeled probe, and 300 ng of sonicated salmon sperm DNA. Binding reactions were preincubated for 10 min at room temperature; labeled probe was added and incubated for an additional 15 min and then resolved by 4% native PAGE in 0.5xTris-Borate-EDTA buffer (2–3 h at 200 V). Gels were dried and exposed to film at −70 °C.

Oligonucleotides were also end-labeled with [γ-32P]ATP using T4 polynucleotide kinase (Boehriger Mannheim, Indianapolis, IN, USA). GST-Purα or GST was preincubated for 30 min at various temperatures as indicated, and subsequently incubated with 54,000 cpm of labeled probe in a final volume of 20 μL for 1 h at 4 °C. Complexes were resolved by electrophoresis in 9% native polyacrylamide gels in 0.5X TBE. Electrophoresis was carried out at 180 V for 3–4 h at 4 °C. After electrophoresis, the gel was dried and exposed to X-ray film.

### 4.10. DNA Helix-Destabilizing Activity of Purα 

Helix-destabilizing assays were performed with the partially ds substrate (in vitro-transcribed srRNAs) and GST or GST-Purα for 60 min at 37 °C in the presence of ATP. 

### 4.11. Unwinding Assay

The substrate utilized in unwinding assays was made by annealing the 15-mer universal sequencing primer (U.S. Biochemical, Cleveland, OH) to srRNA) The primer was labeled with [α32P] dATP using DNA polymerase (Boehringer-Mannheim, Indianapolis, IN, USA). This resulted in single-stranded circular DNA with an 18 bp double-stranded region. The unwinding reaction was carried out by the method described previously [42]. Reactions were stopped by the addition of SDS to a final concentration of 0.3% and EDTA to a final concentration of 0.05 M. Samples were analyzed on 9% native polyacrylamide gels in 0.5X TBE buffer for 3 h at 160 V at room temperature to designate the displaced srRNA and the annealed substrate, respectively.

### 4.12. Nuclease Assay 

The substrate that was utilized in nuclease assays was made by annealing the 500 ng of 22-mer primer to in vitro transcribed 338-base RNA representing the Purα-associated ribosomal RNA. Prior annealing the 22-mer primer was end-labeled with [γ-^32^P]ATP using T4 polynucleotide kinase (Boehriger Mannheim, Indianapolis, IN, USA). This resulted in a single-stranded 338-base RNA with a 22-bp double-stranded region. Nuclease reaction was carried out by the method described previously [73]. Nuclease assays were performed with the partially double-stranded RNA-DNA/RNA-RNA substrate and GST, or GST-Purα for 60 min at 37 °C. Reactions were also performed in the absence of either protein. To study whether the probe was cleaved by Purα, the levels of the displaced [γ-^32^P]-labeled 22-mer oligonucleotide and the annealed substrate, respectively, it was demonstrated by the use of a substrate that was denatured at 72 °C and a substrate incubated at 37 °C. Reactions were stopped by the addition of EDTA to a final concentration of 0.05 M. Samples were resolved by electrophoresis in 6% or 9% native polyacrylamide gels in 0.5X TBE buffer for 3 h at 160 V at room temperature. After electrophoresis, the gel was transferred to Whatman filter paper, dried, exposed to X-ray film, and detected by autoradiography. Quantification was performed by densitometry and analyzed with Adobe Photoshop software, CC 2015 and 2021. 

### 4.13. GST Pull-Down Assays

In vitro protein–protein studies were performed using in vitro translated Purα, or Purα recombinant protein, which was fused to glutathione *S*-Transferase to determine whether it interacts to other proteins and whether this interaction is RNA-mediated. Briefly, 3 μL of [^35^S]-labeled and in vitro-translated Purα was incubated with 5.0 µg of GST or fusion proteins GST-Tata or GST-E2F1 coupled to glutathione Sepharose 4B beads in 300 μL of Lysis Buffer for 2 h at 4 °C with continuous rocking. The beads were then separated by centrifugation and washed five times with lysis buffer. The bound proteins were eluted with Laemmli sample buffer, heated to 95 °C for 10 min, and separated by SDS-PAGE. Tat and E2F1 plasmids were cloned in pGEX-2T prokaryotic expression vector, and Purα was cloned in pcDNA3 expression plasmid, respectively. Purα and Tat were detected by either fluorography. For experiments, Purα, 35S labeled Purα, was pretreated with 10 μg/mL of soluble DNase-free RNase (Boehringer Mannheim, Mannheim, Baden-Wurttemberg, Germany) for 30 min at room temperature before the pull-down assay as described above. In experiments in which it was necessary to selectively remove RNase from the binding reaction, in vitro translated [^35^S]labeled Purα was treated with 10 units of insoluble RNase (RNase conjugated to agarose beads) (Sigma) per microliter of protein for 2 h at room temperature. After treatment with insoluble RNase, the RNA-free [^35^S]labeled Purα-containing extract was separated from the RNase agarose bead conjugate first by microcentrifugation, followed by filtration through a 0.2 μm cellulose acetate filter Spin-X column CoStar (MilliporeSigma, Burlington, MA, USA). In experiments that used RNase-treated samples, all reactions were supplemented with 40 units of RNase inhibitor (Boehringer Mannheim, Mannheim, Baden-Wurttemberg, Germany) and 0.5 mM DTT. For reconstitution experiments, 9 μg of in vitro-transcribed srRNA in the sense or antisense orientation were included in the binding reactions.

### 4.14. Overexpression and Purification of Recombinant Proteins 

GST fusion proteins were expressed in E. coli and purified according to the method described previously [28]. Briefly, bacteria were grown overnight at 37 °C in a Luria Bertani medium supplemented with 100 mg/L ampicillin. After 16 h, bacteria were diluted 1:10 in fresh LB medium, grown to an OD at 595 nm of 0.6, and induced for 2 h at 37 °C with 0.5 mM isopropyl-β-D thiogalactopyranoside (IPTG). Bacteria were collected by centrifugation at 7000× *g* at 4 °C, resuspended in NETN buffer (20 mM Tris, pH 8.0, 100 mM NaCl, 1 mM EDTA, and 0.5% Nonidet P-40) containing 2 mg/mL lysozyme, 1 μg/mL leupeptin, 1 μg aprotinin, and 1 mM phenylmethylsulfonyl fluoride, and sonicated on ice. The bacterial lysate was centrifuged at 40,000× *g* at 4 °C. Glutathione-Sepharose beads (Amersham Pharmacia Biotech, Piscataway, NJ, USA) were added to the supernatant and incubated at 4 °C for 2 h. Beads were pelleted and washed three times with 25–50 volumes of NETN buffer each time. The integrity and purity of the GST fusion proteins were verified by SDS-PAGE followed by Coomassie Blue staining. Known amounts of BSA were included on the same gel for determination of the yield of the full-length proteins. Radiolabeled Purα and Tat proteins were synthesized with the TNT-coupled wheat germ extract system according to the manufacturer’s recommendations (Promega, Madison, WI, USA). BSA was obtained commercially (New England BioLabs, Beverly, MA, USA).

### 4.15. In Vitro Transcription and RNA Binding Assays 

In vitro-transcribed RNA was generated from BamHI linearized pGL3-T7-luciferase according to the recommendations of the manufacturer (Promega, Madison, WI, USA). pGL3-T7 DNA was removed by digestion with DNaseI following the transcription reactions according to the recommendations of the manufacturer (Promega, Madison, WI, USA). RNA was transcribed in vitro by using T7 RNA polymerase. Transcription reactions (20 μL) contained 500 ng of linearized DNA templates, 40 mM Tris⋅HCl (pH 7.5), 6 mM MgCl_2_, 10 mM DTT, 4 mM spermidine, 20 units of RNase inhibitor, 0.5 mM of ATP, GTP, and CTP, 50 μCi of [α^32^P]UTP (400 Ci/mmol), and 20 units of T7 RNA polymerase (Boehringer Mannheim). Reactions were incubated for 1 h at 37 °C followed by the addition of 20 units of RNase-free DNase I and subsequently were incubated for 15 min at 37 °C. Radioactively labeled RNA was further gel-purified on a 9% denaturing polyacrylamide gel. The srRNA-338-base RNA was transcribed in the sense and antisense orientations from *Xba*1 linearize pCDNA vectors containing the cDNA of the 338-base RNA in the sense and antisense orientations, respectively.

In vitro-translated 35S labeled Purα was incubated with GST-Tat protein. Incubation was performed for 1 h at 4 °C, and the resin was subsequently pelleted and washed with binding buffer. Bound proteins were eluted by boiling in Laemmli sample buffer and, after separation by SDS/PAGE, were detected by autoradiography. An amount equivalent to 10% of the material used for the assays was applied to the input lanes. In vitro-translated 35S labeled Purα was incubated with GST-Tat, also in the presence of RNase. For reconstitution of the interaction between Purα and Tat protein by the addition of RNA, in vitro-translated Purα was pretreated with RNase before incubation with Tat with agarose-RNase beads prior its incubation with GST-Tat, in the presence of the in vitro-transcribed srRNA 338-base fragment in the sense and antisense orientation.

### 4.16. DNA Transfections and Preparation of Cell Extracts 

Transient transfections were carried out by the calcium phosphate technique as described [28]. In brief, 2 × 10^5^ neuronal cells were plated on 60 mm dishes and transfections were carried out with 3.5 μg of DNA. The precipitate was removed from the cells 3 h later, and cells were subsequently subjected to a glycerol shock. Forty-eight hours after transfection, cells were lysed for 20 min on ice in LB 150 (50 mM Tris, pH 7.4/150 mM NaCl/5 mM EDTA/0.1% Nonidet P-40) buffer containing 1 μg/mL leupeptin, 1 μg/mL aprotinin, 1 mM PMSF, and 50 mM NaF. Cell debris was pelleted by centrifugation at 14,000 rpm for 15 min at 4 °C. The supernatant was assayed for protein content by Bradford analysis (Bio-Rad) and was used for luciferase assay; experiments were performed with 5 × 10^6^ cells in 60-mm dishes. Cells were harvested 48 h post-transfection, and protein extracts were used to examine the level of luciferase activity [28].

### 4.17. Quantification of Neuronal Cell Injury: Caspase-GLO 3/7 Activity Assay

Apoptosis was assessed for activation of Caspase-3, using the Caspase-Glo™ 3/7 assay kit (Promega, Madison, WI, USA), according to the manufacturer’s instructions. Approximately 1000 neuronal cells were analyzed in a final volume of 100 microliters culture medium per well. One hundred microliters of Glo reagent were added to the culture medium (1:1), then after cell lysis induction, the luminescence was recorded (RLU/s) on a Luminometer (Zylux Corporation, Pittsburgh, PA, USA). Data were analyzed using Excel software (Microsoft 365). 

### 4.18. Luciferase Reactions 

Coupled in vitro transcription/translation reactions were performed with XbaI linearized pGL3-T7-luciferase and various amounts of exogenously added BSA, GST, and GST-Pura using the TNT-coupled transcription/translation wheat germ extract following the manufacturers instructions (Promega, Madison, WI, USA). An equal-volume aliquot of each reaction was assayed for luciferase activity according to the manufacturer’s recommendations (Promega, Madison, WI, USA). Alternatively, in vitro-transcribed luciferase RNA was incubated with various amounts of BSA, GST, or GST-Pura in coupled transcription/translation extract, and subsequently, an equal volume of the reaction was assayed for luciferase activity. Experiments were performed at least in triplicate with qualitatively similar results. Relative light units were converted into fold activity for graphical representation.

### 4.19. Statistical Analysis

Statistical analysis was described previously [31,32,74]. In brief, analysis was performed using SPSS (IBM Corp., released 2017. IBM SPSS Statistics for Windows, Version 25.0. Armonk, NY, USA). All data are represented as the mean ± standard error for all performed repetitions. Means were analyzed by a one-way ANOVA, with Bonferroni correction. Statistical significance was defined as *p* < 0.05. 

Search of database: we used statistical and computational methods, including Human genome database [75,76,77,78,79,80,81,82,83,84,85,86,87,88,89,90]; NCBI PUBMED, BLAST, Entrez Nucleotide database [91]; Bioinformatics statistical analysis were performed using CLUSTAL W multiple sequence alignment program, DNA to Protein Translation [92]; NetGene2 World Wide Web Server, GENSCAN [93,94]; ZiFiT version 3.0, Splice Site Prediction by Neural Network program, and PyMOL software (Academic PyMOL) [95].

## 5. Conclusions

Currently, FASD remains underdiagnosed. In clinical practice, the diagnosis of FASD is usually made postnatally, in part because conventional ultrasound and MRI lack sufficient resolution to detect the morphological features of FASD early in development [96,97], and in part because there are no accepted molecular biomarkers for FASD, even postnatally, although a few studies demonstrated low MBP levels in EtOH-exposed cells [98,99,100]. The present study proposes a novel class of biomarkers based on small ribosomal RNAs that are developmentally regulated and can be detected as early as 9 weeks gestation. The data presented suggest that EtOH inhibits the synthesis of 18s-rRNA, which is formed by RNA cleavage through the action of the nucleic acid-binding protein Purα, which itself requires 18s-rRNA for its synthesis. Purα is a transcription factor that binds to the promoter of the MBP gene, and possibly other genes in the myelin development program. Thus, by inhibiting Purα expression, exposure of the fetus to EtOH results in the dysmyelination that is characteristic of FASD/FAS. The process can be monitored by testing the contents of FB-E isolated non-invasively from maternal blood. Inhibition of expression of both MBP and 18-s rRNA levels in FB-Es have now correlated strongly with an anatomical hallmark of FASD, i.e., reduced eye diameter, in the fetus. Whether either of these biomarkers or others, individually or in combinations, will prove to be good predictors of which at-risk children will develop FASD will await prospective clinical studies. EtOH exposure suppresses expression of srRNAs in the developing brain, which is highly correlated with other molecular markers (e.g., low levels of MBP mRNA and protein, and elevated levels of activated caspase-3) in fetal cell type-specific exosomes. EtOH exposure also increases activity of the pro-apoptotic caspase-3 in human cortical neurons and inhibits the survival of neurons. Exogenous srRNAs reverse the EtOH-mediated down regulation of srRNAs, possibly by stabilizing endogenous srRNAs, or by increasing association of cellular proteins with srRNAs, modifying gene transcription. These findings suggest a potential mechanism for EtOH-mediated neurotoxicity via alterations in 18S rRNA processing. Thus, ribosome biogenesis may be a novel target to ameliorate FASD in utero or after birth. Finally, srRNA as a novel class of non-coding RNA biomarkers could be used as an additional molecular approach in the therapeutic intervention for treatment of FASD. We previously proposed MBP-based molecular approaches, such as enhancing MBP transcription by targeting the affected biomarkers, or their promotors and suppressors, e.g., Purα.

## Figures and Tables

**Figure 1 ijms-24-13714-f001:**
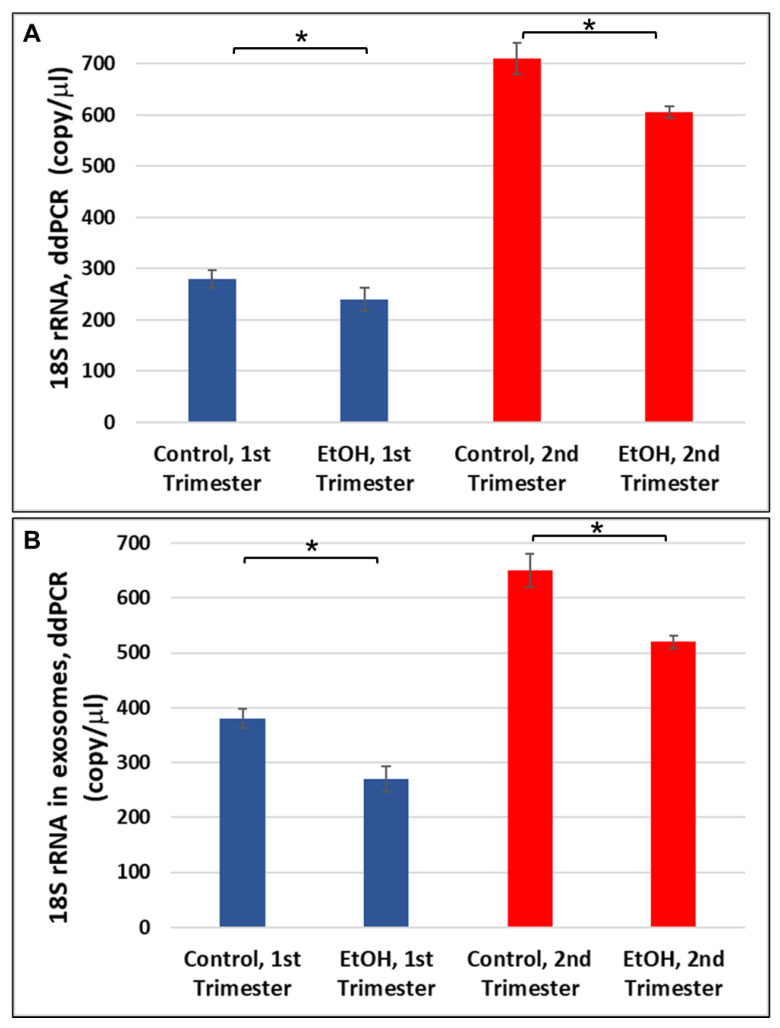
Downregulation of 18S rRNA in FASD. (**A**) Human fetal brain tissues from 1st (*n* = 10) and 2nd (*n* = 10) trimesters who had no EtOH exposure, or with EtOH exposure were measured (in triplicate) for 18S rRNA expression by ddPCR. Downregulation was greatest in EtOH cases. (Means of triplicate assays ± SD). (**B**) Twenty EtOH-exposed FB-Es from maternal blood were compared with twenty individually-matched unexposed controls, ten matched pairs from the 1st trimester, and ten from the 2nd trimester. Each assay was performed in triplicate. Error bars indicate standard deviations based on numbers of fetuses, or FB-Es. Significance levels are for the comparison between all EtOH-exposed and all unexposed controls within the indicated trimester (1st or 2nd), based on ANOVA. * *p* < 0.05.

**Figure 2 ijms-24-13714-f002:**
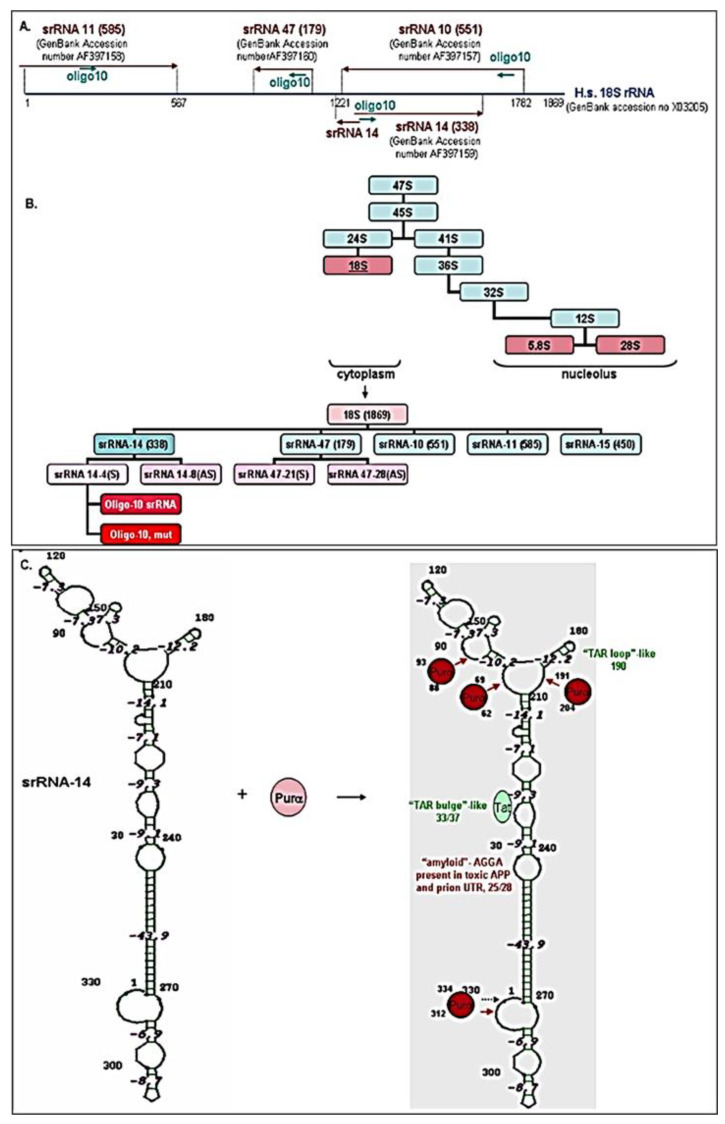
Generation of 18S-homologous small ribosomal RNAs (srRNAs). (**A**) Association of 18S ribosomal RNA with cellular protein (Purα), results in the formation of 4 small RNA molecules. Oligo-10 represents 10-mer GC-rich repat sequence within all 4 isolated srRNAs. (**B**) Cytoplasmic or nuclear distribution of ribosomal RNAs. Cytoplasmic localization of smRNAs and or short oligo srRNA fragments within 18S ribosomal RNA. (**C**) Secondary structure of small ribosomal RNA. Stem-loop organization of srRNA, processed by a cellular protein (Purα).

**Figure 3 ijms-24-13714-f003:**
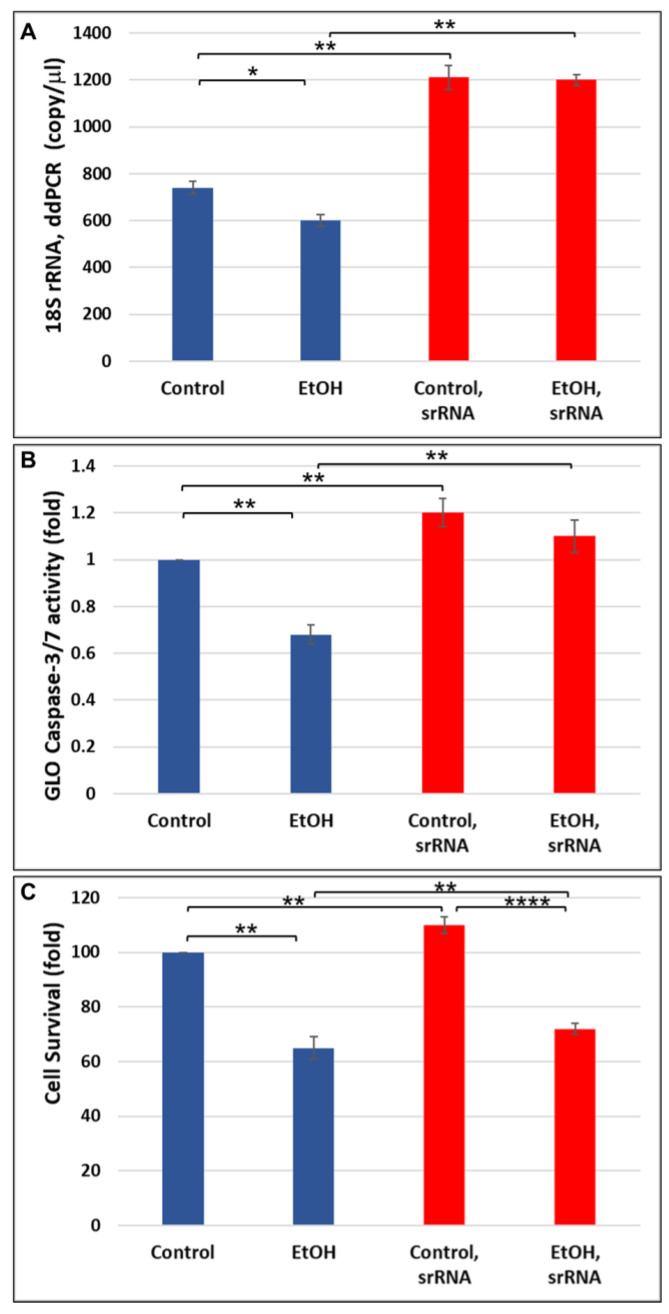
srRNA reverses inhibitory effect of EtOH exposures on 18S rRNA expression in human neuronal cells. (**A**) Neuronal culture was treated with 50 mM of EtOH and incubated with srRNA. 18S rRNA expression was measured by ddPCR (means of triplicate assays ± SD). (**B**) srRNA reverses EtOH-caused Caspase-3 activation, measured by GLO Caspase-3/7 activity assay. Neuronal cultures were treated with 50 mM of EtOH and incubated with srRNA. Caspase-3 activity was measured by GLO Caspase-3/7 assay (means of triplicate assays ± SD). (**C**). srRNA reverses toxic effect of EtOH on cell survival, measured by cell viability assay. Neuronal culture was treated with 50 mM of EtOH and incubated with srRNA. Cell survival was measured by GLO cell viability assay. Error bars are averaged values for the means of triplicate assays ± SD. * *p* < 0.05, ** *p* < 0.01, and **** *p* < 0.0001.

**Figure 4 ijms-24-13714-f004:**
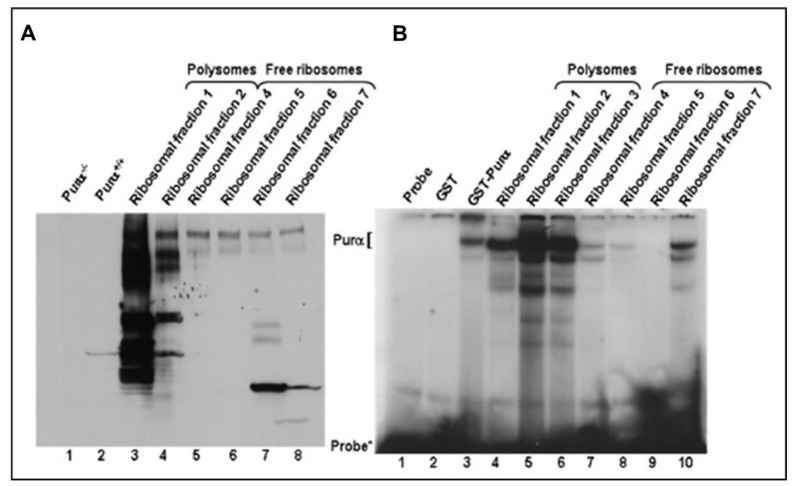
Ribosomal RNA in polyribosomes is in the complex with the cellular protein (Purα). Detection of Purα in ribosomal fractions. (**A**) Ribosome analysis was performed as described in Materials and Methods [38]. Ribosome fractions-1–2, 4–7 (lanes 3–8) were used in electrophoretic mobility shift assays (EMSA) to demonstrate presence of cellular protein Purα in polyribosomal complexes. Lanes 1 and 2 represent cell lysates with (lane 2) or without Purα (lane 1). (**B**) Strong srRNA-Purα association in polysomes (lanes 4–6) compared to free ribosomal fractions (lanes 8–10). Probe (lane 1) was also incubated with GST (negative control, lane 2) or GST-Purα (positive control, lane 3). Lane 7 in left panel could represent partially cleave or degraded forms of Purα. Ribosomal fractions were incubated with srRNA 27-mer GGC-rich oligonucleotide [5′-agcttggaggcgga ggcggcctcggcg-3′] corresponding to the wild-type putative Purα-binding site (GGN repeats), present in srRNA, synthesized, annealed to complementary synthetic oligonucleotide in vitro and radiolabeled with [γ-^32^P]-dCTP using Klenow DNA polymerase (New England Biolabs, Beverly, MA, USA). DNA binding reactions were performed in 20 μL containing 10 μg of the ribosomal fractions, 2 μL of DNA binding buffer, 1 ng of labeled probe. Binding reactions were incubated for 25 min at room temperature, and then resolved by 4% native PAGE in 0.5xTris-Borate-EDTA buffer (2–3 h at 200 V). Gels were dried and exposed to film at −70 °C.

**Figure 5 ijms-24-13714-f005:**
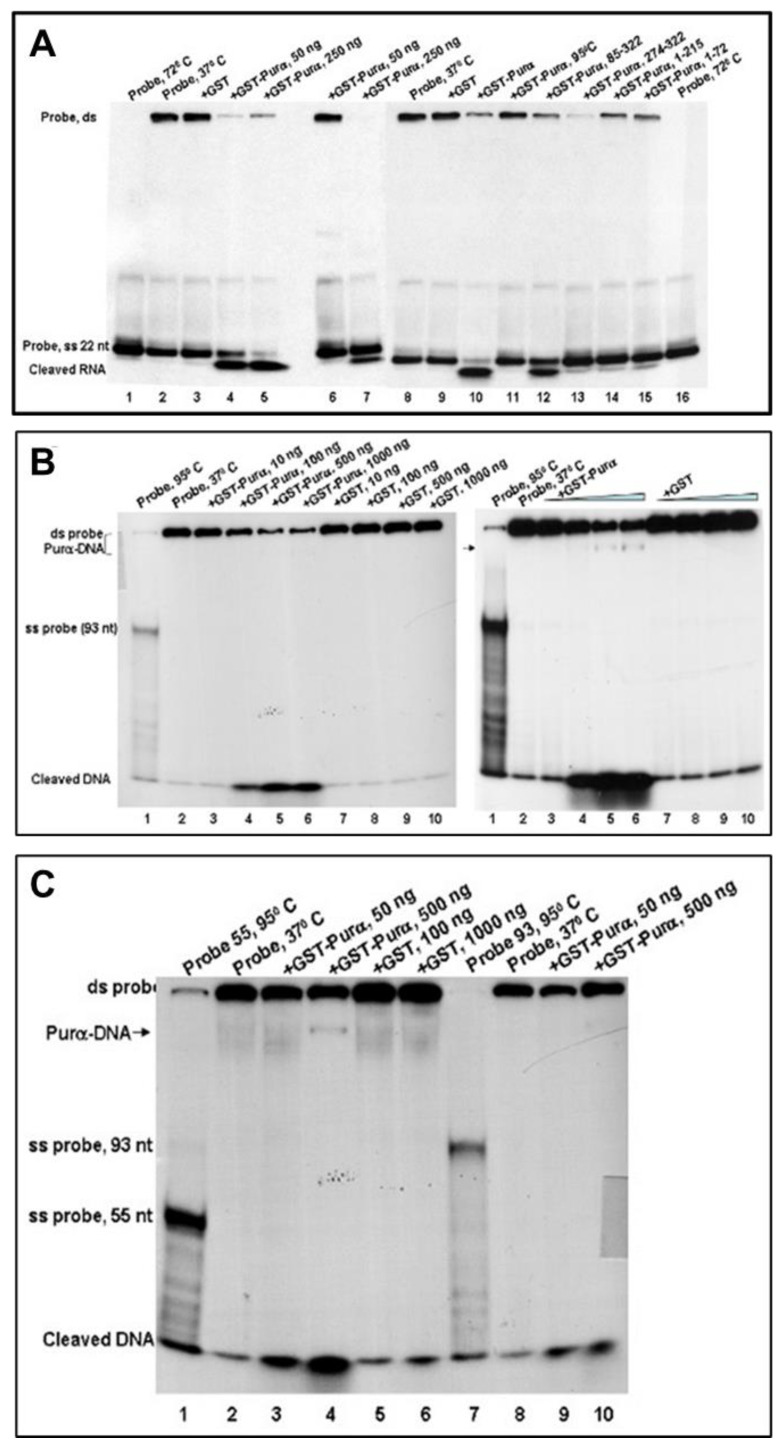
In vitro-transcribed 18S ribosomal RNA is cleaved by a cellular protein (Purα). (**A**) Purα exhibits nuclease activity on 18S ribosomal RNA. Sequence- and dose-specific (lanes 4–5, 6–7) cleavage of in vitro-transcribed srRNA by Pura and its deletion mutants (lanes 12–15). Lane 1 contains substrate in the absence of protein that has been denatured at 72 °C, or a probe at 37 °C. Helix-unwinding and nuclease assays were performed with the partially ds substrate and 50 and 250 ng of GST–Purα. Lanes 4 and 5 represent two independently purified Purα proteins. (**B**) Purα exhibits DNAse activity in dose-dependent manner. Assays for DNA helix-destabilizing and nuclease activity of Purα were performed with the partially ds substrate and 10, 100, 500, or 1000 ng of GST (lanes 7–10) or 50, 500, or 1000 ng of GST-Purα (lanes 3–6) for 60 min at 37 °C. Lane 1 contains a substrate which has been denatured at 95 °C and lane 2 contains a probe incubated at 37 °C, in the absence of either protein. (**C**) Nuclease activity of Purα does not depend on the length of the probe. Helix-destabilizing and nuclease assays were performed with the partially ds substrate and 100 or 1000 ng of GST (lanes 5 and 6) or 50 or 500 ng of GST–Purα (lanes 3–4, 8–9), either in the presence (Lanes 3–6) or the absence (Lanes 7 and 8) of ATP. The displaced 18-mer and the annealed substrateare shown on the left.

**Figure 6 ijms-24-13714-f006:**
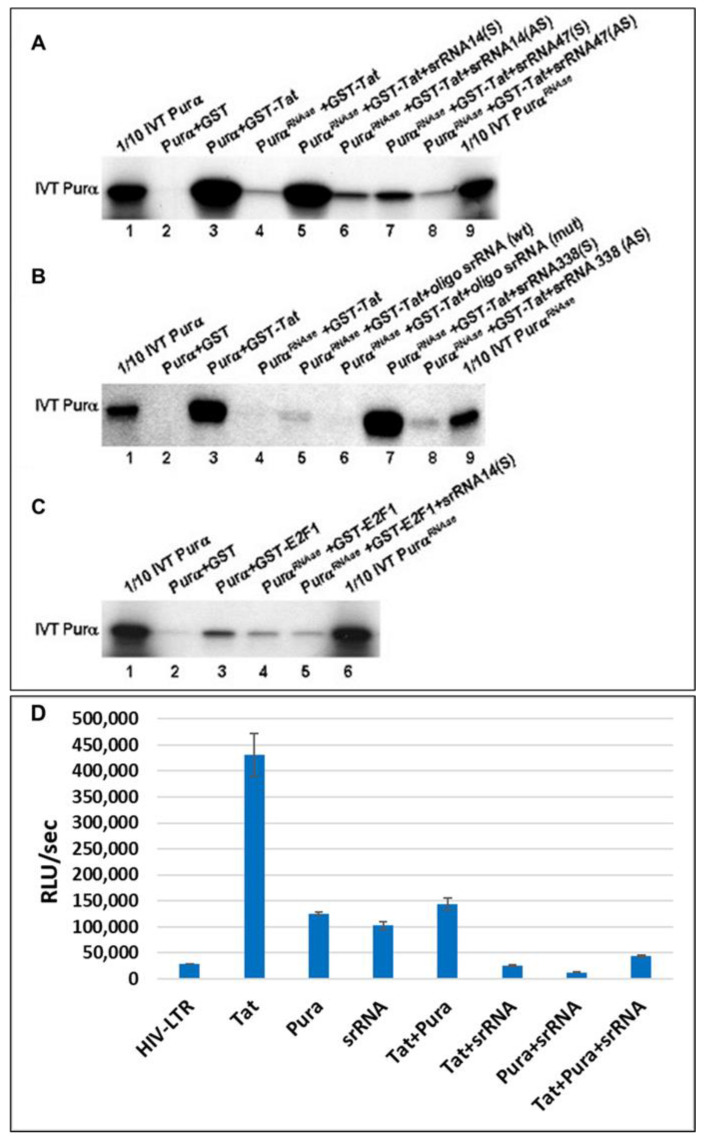
18S homologous srRNA controls host cell–virus association. (**A**) Cellular protein, Purα, and a viral protein, HIV-1 Tat, associate via srRNAs. (**B**) srRNA oligo prevents dissociation of RNase-treated Purα–Tat interaction. (**C**) srRNA oligo is specific for Tat–Purα association. (**D**) 18S homologous srRNA strongly inhibits Tat-activated HIV-1 LTR transcription activity in human neuronal cells (bar 6) and abrogates Pura-induced or Tat- and Pura-induced HIV-1 LTR activation (bars 7–8). In vitro-translated 35S labeled Purα (lane 1) was incubated with GST (lane 2) or GST-Tat protein (lane 3 in (**A**,**B**)) or another cell protein E2F1 (lane 3 in (**C**)) for 1 h at 4 °C. The complex was then pelleted, washed with binding buffer, eluted by boiling in Laemmli sample buffer, and separated by SDS/PAGE. Proteins were detected by autoradiography. An amount equivalent to 10% of the material used for the assays shown in lane 1. 35S labeled in vitro-translated Purα was treated with RNase and then incubated with GST-Tat (lane 4 in (**A**,**B**)), or GST-E2F1 (lane 4 in (**C**)). Reconstitution of the interaction between Purα and Tat protein by the addition of srRNA (lanes 5–8). (**A**) In vitro translated Purα (lane 4) was selectively pretreated with agarose-RNase beads before incubation with Tat, then incubated with GST-Tat in the presence of srRNA-14 (338-nt in sense orientation, lane 5), or with antisense srRNA-14 (338-nt, lane 6), sense srRNA-47 (lane 7), or antisense srRNA-47 (lane 8). Lane 9 contains 1/10 of the amount of 35S labeled Purα used in the binding reactions, pretreated with RNase. (**B**) In vitro-translated Purα after RNase treatment was incubated with GST-Tat in the presence of the synthetic oligo-srRNA representing the consensus GC-rich sequence (CCCGGCCGGU) (lane 5) or its mutant variant (AUGACUUGUC) (lane 6), or with the longer form, sense srRNA-14 (lane 7), or AS srRNA-338 (lane 8). Lane (**C**) Binding reactions with in vitro translated Purα after RNase treatment (as indicated) and GST-E2F, in the presence of the in vitro-transcribed srRNA-338 fragment in the sense orientation (lane 5). (**D**) Functional importance of Purα-associated RNA in Tat and Purα effects on LTR activity. Human primary neuronal cells were transfected with 0.5 μg of LTR-luciferase reporter plasmid in the absence or presence of 1 μg of expression plasmids for Tat, Purα, and the srRNA-338 alone or in combination, as indicated at the bottom of the graph. Luciferase activity was determined 36 h after transfection, and the levels of promoter activation by Tat, Purα, and the srRNA-338 were determined and graphed as fold activation. In all cases, the final amount of DNA in the transfection mixture was brought to 3.5 μg with pCMV DNA. The results are the mean of three independent experiments. Bars indicate standard deviation.

**Figure 7 ijms-24-13714-f007:**
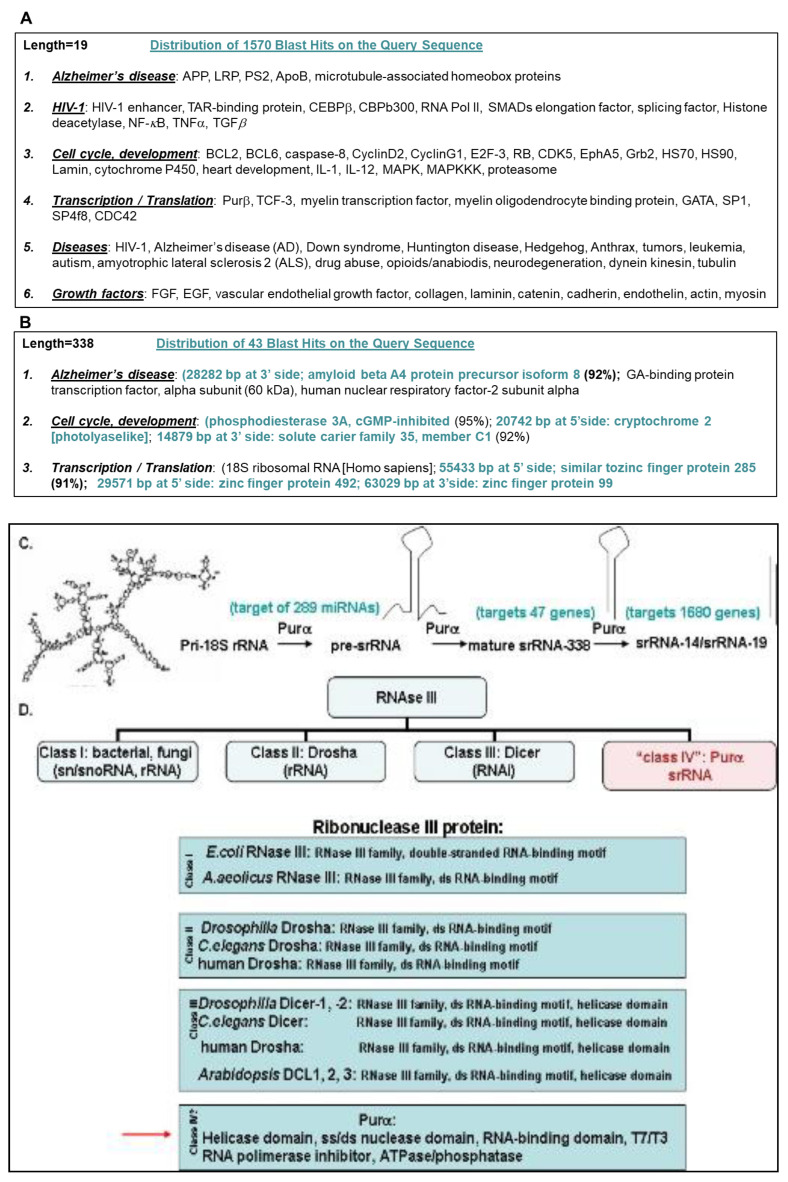
Potential targets of srRNA. (**A**) Blast analysis of srRNAs. Distribution of 1570 genes homologous to 19-nt srRNA. The presence of GGN-rich regions in AD-associated genes was already shown in our previous studies, and evermore, it was demonstrated that srRNA-associated Purα binds to APP promoter and inhibits its transcriptional activity [43]. (**B**) Distribution of 43 genes homologous to 338-nt srRNA. (**C**) Schematic presentation of srRNA processing by cellular proteins. (**D**) Summary for novel activities of srRNA association with a cellular protein Purα, a novel member of Class IV RNAse III. (**E**) Schematic presentation of srRNA-Purα complex activities: protein–protein interaction via RNA, DNA- and RNA-binding, binding to ribosomes, DNA-unwinding, DNase and RNase nuclease activities, phosphatase activity, control of transcription/translation, and involvement in DNA-damage repair. Most Purα interactions are RNA-dependent.

**Figure 8 ijms-24-13714-f008:**
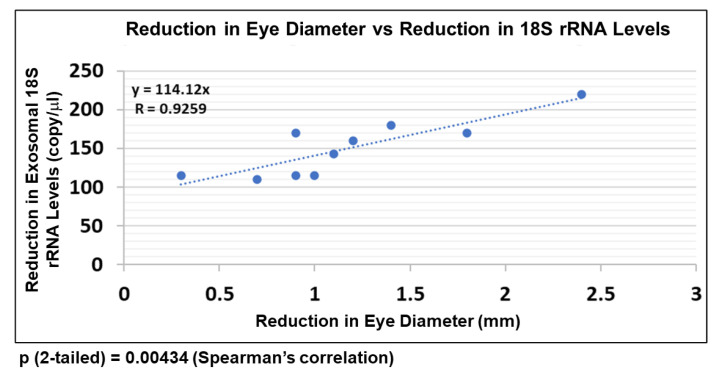
Reduction in FB-E level of 18s-rRNA correlates closely with a reduction in human fetal eye size. Eye diameters were measured in histological sections. rRNA levels were measured by ddPCR (for copy/μL). Ten EtOH-exposed fetal eye tissues each were paired with an age- and sex-matched control, and with their matching maternal blood samples from 1st- and 2nd-trimester pregnancies (*n* = 20 total). Assays were performed in triplicate on contents of FB-Es isolated from the maternal blood. Correlation between reduction in eye size (difference between EtOH exposed fetus and its paired control) and reduction in rRNA levels is presented as a scatter plot. Calculations are based on Spearman’s correlation on exact two-tailed probabilities critical *p* values for N > 2 ≤ 18 [44,45].

**Table 1 ijms-24-13714-t001:** Clinical characteristics of subjects used in the experiments.

	EtOH Consumed Women (*n* = 20)	Controls (No EtOH Users, *n* = 20)
Maternal Age (years ± SD)	28.0 ± 2.7	24.15 ± 2.3
Gestational Age (weeks ± SD)	15.22 ± 1.6	14.92 ± 1.58
Race: White (%)	50	50
Race: Black (%)	50	50
Fetal Sex, Male (%)	50	50
Fetal Sex, Female (%)	50	50

## Data Availability

This study collected demographic, behavioral, and laboratory data from normal healthy women, and from women who drank alcohol during pregnancy. Our research team supports all these activities and has developed a data sharing plan. We also recognize that additional benefits from data sharing may arise in the future that are not apparent at this time, and we are prepared to work specifically with NIH in addressing all requests for raw data. At the present time, we have not deposited any of these raw data in an existing databank, but will make the data available to other investigators on request, in a manner consistent with NIH guidelines. Consistent with NIH policy, shared data will be rendered “free of identifiers that would permit linkages to individual research participants and variables that could lead to deductive disclosure of the identity of individual subjects” Intellectual property and data generated under this project will be administered in accordance with both University and NIH policies, including the NIH Data Sharing Policy and Implementation Guidance of 5 March 2003, and 0925-0001 and 0925-0002 (Rev 07/2022 through 01/31/2026). With this caveat observed, data will be made available to the NIH/NICHD/NIAAA. Sufficient identifiers will be provided to the NIH so that research participants can be assigned a Global Unique Identifier (GUID), which is a universal subject ID that protects personally identifiable information (PII). Using the GUID, NDAR can bring together multiple types of data collected from a single participant, regardless of where and when those data were collected. Biological samples (blood, serum, exosomes and RNAs) and data that are shared will be completely free of identifiers that would permit linkages to individual research participants. We will make biological samples, deidentified data, and associated documentation available to users only under a data sharing agreement that provides for (1) a commitment to using the data only for research purposes; (2) a commitment to securing the data using appropriate computer technology; and (3) a commitment to destroying or returning remaining samples after analyses are completed. Intellectual property and data generated under this project will be administered in accordance with both University and NIH policies, including the NIH Data Sharing Policy and Implementation Guidance of 5 March 2003. As the FAIR data bank receives approval from the NIH, the data will be made available to that group as well. The NIH will be implementing a new specific policy regarding data sharing https://grants.nih.gov/grants/guide/notice-files/NOT-OD-21-014.html, as of 25 January 2023. We will adopt that policy also. Data will be also available at https://www.mdpi.com/ethics.

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
