# Peer review of "Effects of In Utero EtOH Exposure on 18S Ribosomal RNA Processing: Contribution to Fetal Alcohol Spectrum Disorder"

_ijms, 2023, doi:10.3390/ijms241813714_

Round 1
Reviewer 1 Report
Comments to the Authors of manuscript number: ijms-2574323 entitled “Effects of In Utero EtOH Exposure on 18S Ribosomal RNA Processing: Contribution to Fetal Alcohol Spectrum Disorder”.
Fetal alcohol spectrum disorder (FASD) impairs brain development due to alcohol exposure, with unclear mechanisms and genetic influence. The Authors explored EtOH's impact on ribosome formation in fetal brains. The result suggests EtOH's harmful effects on brain development, but introduced RNA could counteract it. This informs FASD mechanisms and proposes early detection through exosomes. Targeting ribosome formation could mitigate FASD effects.
Authors describe the importance of RNA and ribosomal RNA very widely. Even it was mentioned that 18S rRNA is the most stable reference gene for normalizing qRT-PCR data. But it seems that there is lack a clear hypothesis, and it should be clearly indicated why 18S rRNA was chosen.
Moreover, the introduction cannot be finished with the description of what is shown in the paper, but there should be presented the goal and what methods were used to achieve the goal. The sentence e.g. : “We show that fetal exposure to EtOH disrupts ribosome biogenesis, processing of pre[1]ribosomal RNA..” – it is rather the summary of the study.
Although the exposure pattern to alcohol is described, but it is unclear if all subject were exposed at the same level. There is also no information on how the fetal brain samples were collected. There is only mentioned that fetal brain and eye tissues were collected. If Authors collected eye tissue, why it is not mentioned in abstract? What purpose eyes were collected for? It should be explained clearly. If for comparison. Generally, methods are described properly, all procedures are supported by reference. The discussion section of a research paper interprets and analyzes findings in the context of existing knowledge. Conclusions are too long. Authors could also highlight any limitations, such as study design or methodology issues, which can help contextualize the results and suggest avenues for further research. The manuscript is worth to be published after small correction.
Author Response
Fetal alcohol spectrum disorder (FASD) impairs brain development due to alcohol exposure, with unclear mechanisms and genetic influence. The Authors explored EtOH's impact on ribosome formation in fetal brains. The result suggests EtOH's harmful effects on brain development, but introduced RNA could counteract it. This informs FASD mechanisms and proposes early detection through exosomes. Targeting ribosome formation could mitigate FASD effects.
Authors describe the importance of RNA and ribosomal RNA very widely. Even it was mentioned that 18S rRNA is the most stable reference gene for normalizing qRT-PCR data. But it seems that there is lack a clear hypothesis, and it should be clearly indicated why 18S rRNA was chosen.
- An hypothesis is added to the Introduction.
Moreover, the introduction cannot be finished with the description of what is shown in the paper, but there should be presented the goal and what methods were used to achieve the goal. The sentence e.g. : “We show that fetal exposure to EtOH disrupts ribosome biogenesis, processing of pre[1]ribosomal RNA..” – it is rather the summary of the study.
- The Introduction now ends with a statement of the goal and the above mentioned sentence is moved to the Conclusions.
Although the exposure pattern to alcohol is described, but it is unclear if all subject were exposed at the same level.
- Alcohol amounts are added in the Methods Not all subjects were exposed to the same level of alcohol. The total cumulative alcohol dose for EtOH-consuming mothers, ranged from 57–168 drinks (or 12–30 drinks/month) in the 1st trimester, and from 54.4 to 1827.5 drinks (or 6–320 drinks/month) for 2nd trimester. A drink was estimated as the equivalent of one shot (1.5 oz of brandy or 5 oz of wine (Darbinian et al, 2023). This is now detailed in Methods, Assessment of Alcohol Exposure in Pregnancy:
There is also no information on how the fetal brain samples were collected. There is only mentioned that fetal brain and eye tissues were collected.
- The Methods section Sample Collection and Processing has been expanded, with more details given about how the tissue was collected and characterized histologically.
If Authors collected eye tissue, why it is not mentioned in abstract? What purpose eyes were collected for? It should be explained clearly. If for comparison.
- Yes, we also collected eye tissues, and added a new Figure 8 to show the correlation of biomarkers (srRNA) in brain-derived exosomes with reliable, easily measured facial markers for FASD, i.e., small eyes.
Generally, methods are described properly, all procedures are supported by reference.
- Methods have now more details.
The discussion section of a research paper interprets and analyzes findings in the context of existing knowledge. Conclusions are too long.
- The Discussion section is edited. Conclusions are bulleted now.
Authors could also highlight any limitations, such as study design or methodology issues, which can help contextualize the results and suggest avenues for further research.
- A limitations paragraph is added now in Discussion.
The manuscript is worth to be published after small correction.
- Thank you.

Reviewer 2 Report
This paper “Effects of In Utero EtOH Exposure on 18S Ribosomal RNA Processing: Contribution to Fetal Alcohol Spectrum Disorder” investigated whether fetal exposure to EtOH disrupts ribosome biogenesis and processing of pre-ribosomal RNAs and ribosome assembly. The article presents sufficient background investigation with reasonable data analysis, which are in line with the readers’ interest of International Journal of Molecular Sciences. However, there are still some shortcomings that need to be further improved or explained.
Comments:
Q1. The resolution of Figure 2 is too low to distinguish many annotations in the figure?
Q2. The background of this article is comprehensive, however, the introduction lacks paragraph divisions. Additionally, how can authors effectively showcase the primary innovation discussed in this article?
Q3. The content of the article is sufficient, but the format is seriously inconsistent with the requirements of the journal.
Q4. Figure 4, What does the relationship between molecular weight of Polysomes and Free ribosomes?
Q5. Figure 5, What's the molecular weight of cleaved DNA? Based on the results in the figure, it can be seen that the DNA is completely hydrolyzed, how can the authors guarantee that.
Q6. How did the authors control the amount of protein loading in each lane? According to the results in the figure, it is easy to see that the amount of protein loading varies greatly, and whether this will affect the result analysis.
Q7. Markers were missing from the results of electrophoresis experiments.
Q8. The conclusion section should be restructured to serve as an effective summary of this article, devoid of discussions through references.
Author Response
Q1. The resolution of Figure 2 is too low to distinguish many annotations in the figure?
A1. Resolution of Figure 2 has been improved.
Q2. The background of this article is comprehensive, however, the introduction lacks paragraph divisions. Additionally, how can authors effectively showcase the primary innovation discussed in this article?
A2. Paragraph divisions have been added in the Introduction. The rationale for focus on rRNA has been added and the novelty of this study (the identification and changes of a novel class of small 18S-homologous ribosomal RNAs by prenatal alcohol exposure) is now highlighted.
Q3. The content of the article is sufficient, but the format is seriously inconsistent with the requirements of the journal.
A3. The format is edited now according to the Journal’s requirements, including the Introduction and Conclusion.
Q4. Figure 4, What does the relationship between molecular weight of Polysomes and Free ribosomes?
A4. Initially, we looked at the molecular weight of free Pura (41 kDa), or high molecular weight complexes of Pura in Polysomes or free ribosomes since a probe used in these studies for Figure 4 had specific GC-rich Pura binding sites. Polysomes (i.e., the number of ribosomes bound to a single mRNA molecule)usually are ensembles of two or more consecutive ribosomes that translate mRNA into proteins Our densitometry studies indicated that the relationships between polysomes and free ribosomes are at least 2-fold – and reached up to 10-fold in some samples.
Q5. Figure 5, What's the molecular weight of cleaved DNA? Based on the results in the figure, it can be seen that the DNA is completely hydrolyzed, how can the authors guarantee that.
A5. Our hypothesis is that Pura first unwinds 2 strands of DNA, and then cleaves 1-2 nucleotides from one strand, since as seen in Figure 5A, the cleaved RNA or DNA is smaller than that of a probe (22 nt in Figure 5A; or 93nt in Figure 5B, lanes 4-6; or 93nt or 55-nt in Figure 5C).
Q6. How did the authors control the amount of protein loading in each lane? According to the results in the figure, it is easy to see that the amount of protein loading varies greatly, and whether this will affect the result analysis.
A6. All used proteins, GST or GST-Pura were recombinant proteins, whose amounts were measured using routine Bio-Rad protein spectrophotometry and confirmed by SDS-gel electrophoresis.
Q7. Markers were missing from the results of electrophoresis experiments.
A7. Since all electrophoresis experiments were radiolabeled, and we used radiolabeled probes of certain sizes that acted as markers.
Q8. The conclusion section should be restructured to serve as an effective summary of this article, devoid of discussions through references.
A8. Thank you, Conclusion is restructured.

Round 2
Reviewer 2 Report
Thanks for these modifications, I have no any other comments.